# Inter- and intra-event rainfall partitioning dynamics of two typical xerophytic shrubs in the Loess Plateau of China

Jinxia An[1,2], Guangyao Gao[1,2,5], Chuan Yuan[3], Juan Pinos[4], Bojie Fu[1,2,5]

[1]State Key Laboratory of Urban and Regional Ecology, Research Center for Eco-Environmental Sciences, Chinese Academy of Sciences, Beijing 100085, China

[2]University of Chinese Academy of Sciences, Beijing 100049, China

[3]State Key Laboratory of Subtropical Silviculture, Zhejiang A&F University, Hangzhou 311300, China

[4]Surface Hydrology and Erosion Group, Institute of Environmental Assessment and Water Research (IDAEA-CSIC), Barcelona 08034, Spain

[5]National Observation and Research Station of Earth Critical Zone on the Loess Plateau in Shaanxi, Xi'an 710061, China

*Correspondence to*: Guangyao Gao (gygao@rcees.ac.cn)

**Abstract.** Rainfall is known as the main water replenishment in dryland ecosystem, and rainfall partitioning by vegetation reshapes the spatial and temporal distribution patterns of

rainwater entry into the soil. The dynamics of rainfall partitioning have been extensively studied at the inter-event scale, yet very few studies have explored its finer intra-event dynamics and the relating driving factors for shrubs. Here, we conducted a concurrent in-depth investigation of all rainfall partitioning components at inter- and intra-event scales for two typical xerophytic shrubs (*Caragana korshinskii* and *Salix psammophila*) in the

Liudaogou catchment of the Loess Plateau, China. The event throughfall (TF), stemflow (SF), and interception loss (IC) and their temporal variations within the rainfall event as well as the meteorological factors and vegetation characteristics were systematically measured during the 2014-2015 rainy seasons. Our results showed that *C. korshinskii* had significantly higher SF percentage (9.2%) and lower IC percentage (21.4%) compared to *S. psammophila* (3.8%

and 29.5%, respectively), but their TF percentages were not significantly different (69.4% vs. 66.7%). At the intra-event scale, TF and SF of *S. psammophila* was initiated (0.1 vs. 0.3 h and 0.7 vs. 0.8 h) and peaked (1.8 vs. 2.0 h and 2.1 vs. 2.2 h) more quickly, and TF of *S. psammophila* lasted longer (5.2 vs. 4.8 h), delivered more intensely (4.3 vs. 3.8 $mm \cdot h^{-1}$), whereas SF of *C. korshinskii* lasted longer (4.6 vs. 4.1 h), delivered more intensely (753.8 vs.

471.2 $mm \cdot h^{-1}$). For both shrubs, rainfall amount was the most significant factor influencing inter-event rainfall partitioning, and rainfall intensity and duration controlled the intra-event TF and SF variables. The *C. korshinskii* with larger branch angle, more small branches and smaller canopy area, has an advantage to produce stemflow more efficiently over *S. psammophila*. The *S. psammophila* has lower canopy water storage capacity to generate and

peak throughfall and stemflow earlier, and it has larger aboveground biomass and total canopy water storage of individual plant to produce higher interception loss compared to *C. korshinskii*. These findings contribute to the fine characterization of shrub-dominated eco-hydrological processes, and improve the accuracy of water balance estimation in dryland ecosystem.

# 1 Introduction

Rainfall is known as the main replenishment of water resources in arid and semi-arid areas, and water resource is the key factor limiting the function of arid ecosystems (Chesson et al., 2004; Cayuela et al., 2018; Magliano et al., 2019a). Before entering into soil, rainfall is redistributed by plant canopies into throughfall (TF, diffuse water input), stemflow (SF, point water input), and interception loss (IC, evaporation). The sum of TF and SF is defined as "net rainfall". Differences in the distribution of net rainfall caused by plant canopy interception alter the spatial and temporal patterns of rainfall entry into the soil (Martinez-Meza and Whitford, 1996; Li et al., 2009; Van Stan II et al., 2020), and further profoundly affect the water use efficiency of vegetation and ecosystem sustainability (Xu and Li, 2006; Lacombe et al., 2018; Molina et al., 2019). In addition, net rainfall could regulate vegetation physiological metabolic processes through nutrient enrichment (Levia and Frost, 2003; Zhang et al., 2016; Van Stan II et al., 2017; Tonello et al., 2021), ultimately affecting the carbon balance of ecosystems (Chu et al., 2018; Jia et al., 2016). In light of the important role of rainfall partitioning in regulating soil moisture and vegetation patch pattern, investigations of the rainfall partitioning dynamics are imperative for a better understanding of the soil-water-vegetation relationships (Molina et al., 2019; Van Stan II et al., 2020; Zhang et al., 2021a).

Studies on rainfall partitioning have been broadly carried out in different climatic zones and various types of vegetation (Gordon et al. 2020; Rivera and Van Stan II, 2020; Zhang et al., 2021b; Yue et al., 2021). Based on a comprehensive global synthesis, Yue et al. (2021)

found that most TF and SF observations were measured in forests ($n$ = 718 and $n$ = 816, respectively), and that in shrublands was scarce ($n$ = 43 and $n$ = 63, respectively), which was mainly due to that the shrubs have multiple branches and the rainfall partitioning of shrubs is

difficult to be measured compared to forests. Shrubs are the dominant vegetation type in drylands, forming fertile islands by intercepting water and trapping sediments, thus providing important ecosystem goods and services (Levia and Frost, 2003; Llorens and Domingo, 2007; Soulsby et al., 2017). However, the lack of information on the detailed dynamics of rainfall partitioning processes induced by shrubs due to limited studies hinders us form a clear

understanding of shrubs' eco-hydrological role in shaping and sustaining drylands.

Most of the existing studies on the rainfall partitioning by shrub are based on the inter-event scale (Garcia-Estringana et al., 2010; Magliano et al., 2019a). Magliano et al. (2019a) synthesized that for 27 shrub species in drylands, the mean event-based SF%, TF%, and IC% were 9.4%, 63.0% and 27.6%, respectively. Rainfall partitioning by shrubs has been

reported to be determined by biotic and abiotic factors, such as rainfall characteristics (Levia and Frost, 2003; Magliano et al., 2019b) and canopy structure characteristics (Martinez-Meza and Whitford, 1996; Garcia-Estringana et al., 2010; Yue et al., 2021). Take the later for example, vegetation with smooth barks, more branches and vertical branching had advantages on SF generation (Honda et al., 2015; Magliano et al., 2019a; Whitworth-Hulse et

al., 2020b), and a simple vegetation structure and low canopy density are generally corresponding to a relatively high TF rate and low IC rate (Soulsby et al., 2017; Yue et al., 2021). The complexity of shrub structure poses challenges to understand the causes of rainfall

partitioning dynamics under different meteorological conditions, and it is necessary to substantially explore the differences of rainfall partitioning dynamics and main influencing

factors among different shrub species (Levia et al., 2010; Sadeghi et al., 2020).

In addition to the inter-event studies, a few intra-event scale studies have also been reported, which is essential for better understanding of soil moisture distribution and the hydrological cycle in arid regions (Levia et al., 2010; Levia and Germer, 2015; Cayuela et al., 2018; Zhang et al., 2021a). For instance, Zhang et al. (2018) investigated the spatial-temporal

pattern of TF of *C. korshinskii* in arid area of northern China at high temporal resolution (10-min intervals), and they found that temporal heterogeneity of rainfall clearly affected the spatiotemporal dynamics of TF beneath shrub canopies and the wind directions were the main factor affecting TF in different radial directions. Yuan et al. (2019) described the branch SF variability of *C. korshinskii* and *S. psammophil*, and they showed that intra-event branch SF

variability of xerophytic shrubs temporally depended on rainfall characteristics, with longer lag times and greater rainfall amount required to initiate branch SF for *C. korshinskii* compared to *S. psammophila*. It can be found that those studies on temporal dynamics of shrub rainfall partitioning only explored the single-element process (i.e., throughfall or stemflow), and ignored interception loss. Concurrent investigation on all rainfall partitioning

components and the associated influencing factors at the intra-event scale has rarely been reported. Furthermore, to the best of author's knowledge, no previous studies have been reported so far that simultaneously analyzed TF, SF and IC on an intra-event scale in any shrub species. Therefore, a detailed understanding of shrub rainfall partitioning dynamics at

the intra-event scale with high resolution data is sorely needed to improve mechanistic

understanding of shrubs' eco-hydrological role in shaping and sustaining drylands.

This study was designed at the event and process scales to investigate inter-event and

intra-event rainfall partitioning variability, based on field measurements on two dominant

xerophytic shrubs (*C. korshinskii* and *S. psammophila*) during the rainy seasons of 2014-2015

in the Loess Plateau of China. This study integrated the inter-event and intra-event dynamics

of rainfall partitioning by combining TF, SF and IC at the individual plant scale. We mainly

seek to (a) compare the dynamic processes of rainfall partitioning between the two shrubs at

both inter-event and intra-event scales, and (b) elucidate the effects of rainfall characteristics

and vegetation structure characteristics on rainfall partitioning at both scales. Such an

improvement in our understanding of the fine-scale mechanism of rainfall partitioning would

offer valuable insights regarding shrub-water interactions.

## 2    Materials and methods

### 2.1 Site description and experimental design

This study was carried out in the Liudaogou catchment (110°21′-110°23′ E, 38°46′-38°5′ N)

in Shenmu county, Shaanxi Province of China (Fig. 1a). The Liudaogou catchment (6.9 km$^2$,

altitude from 1094 to 1273 m) is located between the northern Loess Plateau and the south

fringe of Mu Us sandy land in North China. This region is characterized by a moderate

temperate continental climate with well-defined rainy and dry seasons. The mean annual

rainfall is 437 mm ranging between 109-891 mm (1971-2013), and the potential evaporation

is 1337 mm yr$^{-1}$ (Jia et al., 2013). Approximately 70-80% of the rainfall events are

concentrated in the warm months between July and September and most of them occur in the

form of torrential rain (Yang et al., 2019). The Liudaogou catchment was characterized by the

natural arid scrub steppe before it was artificially vegetated in the past 20 or 30 years for soil

and water conservation, windbreak and sand fixation. The main land use types include

artificial grassland, artificial shrub and farmland, and the main vegetation species are *Stipa*

*bungeana*, *C. korshinskii* and *S. psammophila*, which are widely distributed in the arid and

semiarid areas of northwestern China (Yuan et al., 2019). The shrub is distributed sparsely

with distinct interspaces. The actual ground cover fraction covered by shrub canopies was less

than 20%, with the rest of soil being directly exposed to rainfall.

    Two representative xerophytic shrubs, *C. korshinskii* and *S. psammophil*a with 20 years

old, were used for the study. Both species are multiple-stemmed deciduous perennial shrubs

with inverted cone crowns and without trunks. They have minimal nutrient requirements,

extensive adaptability and strong stress resistance, which makes them superior in adapting to

resource-poor environments. According to the documentation of *Flora of China* (Chao and

Gong, 1999; Liu et al., 2010) and field observations, the *S. psammophila* has an odd number

of strip-shaped leaves with 2-4 mm in width and 40-80 mm in length, and the *C. korshinskii*

has pinnate compound leaves arranged opposite or sub-opposite with 6-10 cm in length, and

each pinna has 5 to 8 pairs of ovate leaflets (7-8 mm in length and 2-5 mm in width). We

established two plots (one for *C. korshinskii* and the other for *S. psammophila*) at the

southwestern catchment for field observation (Fig. 1a). The two plots share similar stand

conditions, with the sizes of 3294 m$^2$ and 4056 m$^2$, elevations of 1179 m and 1207 m, and

slopes of $13^o$ and $18^o$, respectively. The distance between the two plots do not exceed 1.5 km.

**2.2 Field measurements**

**2.2.1 Measurements of rainfall and meteorological factors**

This study focused on the individual shrub rainfall partitioning of *C. korshinskii* and *S.*

*psammophila* during the 2014-2015 rainy seasons. Gross rainfall was measured using one

tipping-bucket rain gauge with a 0.2 mm resolution (TBRG, with 186.3 $cm^2$ collection area)

(Onset® RG3-M, Onset Computer Corp., USA) in an open area (Fig. 1b). The rainfall

characteristics, e.g., rainfall amount (RA, mm), rainfall duration (RD, h), rainfall interval (RI,

h), average rainfall intensity (I, mm·$h^{-1}$), rainfall intensity at 10-min interval (i.e., 0-10 min,

10-20 min,….) since the start of rainfall ($I_{10}$, mm·$h^{-1}$) were calculated accordingly. For $I_{10}$, the

one after the onset of rainfall is defined as $I_{10\_b}$ (mm·$h^{-1}$), i.e., the rainfall intensity in the first

10 min. The maximum $I_{10}$ during the rainfall process is defined as $I_{10\_max}$ (mm·$h^{-1}$). As the

TBRG has a resolution of 0.2 mm, we define a single rainfall event as one that is greater than

0.2 mm and not raining for at least 4 hours apart (Iida et al., 2012). A meteorological station

was set up at the experimental plot to record wind speed (WS, m·$s^{-1}$) and wind direction (WD,

$^o$) (Model 03002, R. M. Young Company, Traverse City, Michigan, USA), air temperature (T,

$^o$C) and relative humidity (H, %) (Model HMP 155, Vaisala, Helsinki, Finland). Data were

measured every 30 s and averaged at 10 min interval by the data logger (Model CR1000,

Campbell Scientific, Inc., USA).

**2.2.2 Measurements of vegetation characteristics**

Three representative shrub plants with similar crown heights and crown areas were selected

in each shrub species (Table 1). Based on plot investigation, the vegetation traits at the scale of single plant and branch were measured. For each plant, we measured shrub height (SH, m) with a graduated telescopic stick, counted the number of branches (NB), and calculated the projected canopy area (CA, m$^2$) by measuring canopy diameter following the south-north and east-west direction. The total number of branches was 143 and 218 for selected *C. korshinskii* and *S. psammophila* plants, respectively. For each branch, we measured branch length (BL, cm) with a measuring tape, branch angle (BA, $^o$) with a pocket geologic compass, and branch diameter (BD, mm) with a vernier caliper to calculate the total basal area of the shrub (TBA, m$^2$). Thus, four BD categories (0-10, 10-15, 15-20 and > 20 mm) were defined to ensure the appropriate branch amounts within each category. The measured vegetation traits of *C. korshinskii* and *S. psammophila* plants are shown in Table 1.

Water storage capacity of the canopy is a key factor in determining the amount of interception loss (Levia and Herwitz, 2005; Garcia-Estringana et al., 2010) and SF yield (Van Stan II et al., 2020). We selected 10 representative branches for each shrub species outside the stands, to determine the canopy water storage capacity (C, mL/g) using water immersion method, which was widely used in previous studies (Garcia-Estringana et al., 2010; Wang et al., 2012). The C was calculated as the difference between saturated weight and fresh weight divided by the dry biomass of the selected branch. The *C. korshinskii* and *S. psammophila* had a C of 0.85 mL/g and 0.38 mL/g, respectively. In addition, we estimated the total dry aboveground biomass of single plant (TB) for each species according to the allometric growth model developed by Yuan et al. (2016 and 2017) in the same study area. The

allometric growth model had very high accuracy with $R^2$ more than 0.92. The total canopy

water storage of single plant ($C_m$ = TB times C) was calculated to represent the amount of

rainfall absorbed by the shrub canopy during the rainfall event (Table 1).

**2.2.3 Measurements of inter-event rainfall partitioning**

Manual rain gauges (314.12 cm$^2$ collection area) were used to measure event TF at eight

radial directions (E, SE, S, SW, W, NW, N, NE) beneath each shrub canopy (Fig. 1b). For *C.*

*korshinskii*, eight TF gauges were placed under each *C. korshinskii* plant with 50 cm distance

from the base of stems in the eight directions. For *S. psammophila*, twenty TF gauges were

placed under each plant, with twelve of them placed in 50 cm, 100 cm, and 150 cm distances

from the base of stems in four directions (E, S, W, and N), and eight of them placed in 75 cm

and 125 cm distances in the other four directions (SE, SW, NW, and NE). If the rainfall ended

during the daytime, we completed the collection of TF samples within two hours after the end

of rainfall. If the rainfall ended at night, we completed the collection of samples as early as

possible in the next day to minimize evaporation.

A total of 53 branches of *C. korshinskii* (17, 21, 7, 8 for BD categories of 0-10, 10-15,

15-20 and > 20 mm, respectively) and 98 branches of *S. psammophila* (20, 30, 20 and 28 for

BD categories of 0-10, 10-15, 15-20, and > 20 mm, respectively) were used to determine SF

yield, which covered different types of branches. Funnels constructing of flexible aluminum

foil plates were used to collect SF (Fig. 1b). The funnel was fixed to each branch near the

base and sealed with neutral silicone caulk, and a 0.5 cm diameter PVC hose was attached

vertically to transport SF from the funnel to a container with a lid (SF gauges) with minimum

travel time.

### 2.2.4 Measurements of intra-event rainfall partitioning

Among the selected plants, one *C. korshinskii* and one *S. psammophila* plant were selected to record the volume and timing of TF and SF with TBRGs at intra-event scale. A TBRG was installed in each of four radial directions (E, S, W, N) beneath the shrub canopy of each species, to measure the temporal variations of TF within the rainfall event (Fig. 1b). To characterize intra-event SF dynamics, six representative branches of different BD categories were selected for each species, using the following selection criteria: no crossover between the experimental branch and adjacent branches, no inflection point from the tip to the base of the branch, and accessible for easy installation and measurement. These branches were distributed across the four BD categories (0-10, 10-15, 15-20, and > 20 mm, respectively). SF TBRGs were covered with the polyethylene films to prevent the accessing of throughfall and splash (Fig. 1b). Unfortunately, some TBRGs lost a substantial amount of stemflow data and were therefore discarded from the analysis. Four branches were finally identified for each species, located in each of the four BD categories to measure intra-event SF (6.7, 13.5, 18.6, and 22.1 mm for *C. korshinskii* and 7.2, 14.4, 18.2, and 31.3 mm for *S. psammophila*).

### 2.3 Rainfall partitioning calculations

### 2.3.1 Inter-event rainfall partitioning calculations

For each individual shrub, we measured TF volume for each TF gauges, averaged them, and then converted the volume into TF depth ($TF_d$, mm) at each rainfall event. And the percentage of TF (TF%, %) was calculated by dividing $TF_d$ by the RA, and the average TF intensity (TFI,

mm·h$^{-1}$) was calculated by dividing TF$_d$ by the TF duration (TFD, h). The TFD was recorded

by TF TBRGs.

The inter-event SF yield was defined as the total SF volume of a single plant in a rainfall

event. The SF volumes measured on the selected branches were averaged to obtain the

average volume of SF on the branch scale, which multiply the number of branches to obtain

the total SF volume from the plant. The shrub-scale SF equivalent water depth (SF$_d$, mm) and

the average SF intensity (SFI, mm·h$^{-1}$) were calculated. The percentage of SF (SF%, %) was

converted by dividing SF$_d$ by the RA. The SF$_d$ and SFI were calculated by the following

equations (Hanchi and Rapp, 1997; Levia and Germer, 2015):

$$SF_d = (\overline{SF_b} \times n)/(1000 \times CA) \tag{1}$$

$$SFI = (\overline{SF_b} \times n)/(10 \times TBA \times SFD) \tag{2}$$

where $\overline{SF_b}$ (ml) is the average volume of SF on the branch scale, $n$ is the number of

branches of individual plant, CA (m$^2$) is the canopy area of individual plant, TBA (cm$^2$) is the

total basal area of individual plant, and SFD is SF duration (h) recorded by SF TBRGs. The

parameters 1000 and 10 are the unit conversion factor.

The IC depth (IC$_d$, mm) and percentage of IC (IC%, %) were estimated as:

$$IC_d = RA\text{-}TF_d\text{-}SF_d \tag{3}$$

$$IC\% = 100\%\text{-}TF\%\text{-}SF\% \tag{4}$$

The above inter-event rainfall partitioning variables and their explanations are

summarized in Table 2.

## 2.3.2 Intra-event rainfall partitioning calculations

The TF and SF volume and timing within rainfall event were automatically recorded at dynamic intervals between neighboring TBRG tips (0.2 mm). To better reflect fluctuations in rainfall partitioning components, the intra-event TF and SF data were aggregated every 10 minutes to match with the recording interval of gross rainfall. The four TF depths recorded by TBRGs were averaged to obtain the average TF depth at 10-min interval ($TF_{d10}$, mm). The TF intensity at 10-min interval ($TFI_{10}$, mm·h$^{-1}$) was calculated by dividing the $TF_{d10}$ by the 10 min. Meanwhile, SF depth ($SF_{d10}$, mm) and SF intensity at 10-min interval ($SFI_{10}$, mm·h$^{-1}$) were calculated as:

$$SF_{d10} = \sum_{j=1}^{4}\left(186.3 \times SF_{RG,j} \times n_j\right)\Big/(100 \times CA) \tag{5}$$

$$SFI_{10} = \sum_{j=1}^{4}\left(186.3 \times SF_{RG,j} \times n_j\right)\Big/(TBA \times 1/6) \tag{6}$$

where $SF_{RG,j}$ (mm) is the SF depth of the selected $j$th branch category recorded by TBRG at 10-min interval (1/6 h), $n_j$ is the number of branches in the $j$th category of single plant, 4 is the number of BD category (0-10, 10-15, 15-20, and > 20 mm), and 186.3 (cm$^2$) is the collection area of TBRG. The product of $SF_{RG,j}$ and 186.3 is the SF volume from the branch. The parameter 100 is the unit conversion factor.

Based on the calculated $TFI_{10}$ and $SFI_{10}$, the maximum TF and SF intensity at 10-min interval ($TFI_{10\_max}$ and $SFI_{10\_max}$, respectively, mm·h$^{-1}$) of each rainfall event can be determined. The descriptive variables for the intra-event rainfall partitioning also include the lag times of TF or SF corresponding to the rainfall event. Based on the temporal data

recorded by TBRGs (between neighboring tips), the following variables were calculated:

$LG_{TF}$ and $LG_{SF}$ (h), the time lag of TF and SF generation after the start of rainfall,

respectively; $LM_{TF}$, $LM_{SF}$ and $LM_{R}$ (h), the time lag of $TFI_{10\_max}$, $SFI_{10\_max}$ and $I_{10\_max}$ relative

to the onset of rainfall, respectively; and $LE_{TF}$ and $LE_{SF}$ (h), the time lag of TF and SF ending

after the end of rainfall. The intra-event rainfall partitioning variables and their explanations

are summarized in Table 2.

**2.4 Statistical analysis**

Independent-samples T-tests were used to analyze differences in rainfall partitioning

parameters between *C. korshinskii* and *S. psammophila* at both inter-event and intra-event

scales. To detect the effects of meteorological factors on rainfall partitioning, Pearson

correlation analysis was used to test the significance between rainfall partitioning parameters

and meteorological factors at the two scales. The significant correlated factors were

double-checked by partial correlation analysis to determine their individual effects on rainfall

partitioning components. Stepwise regression of these indicators was performed by analytical

tests at the 0.05 level of significance to select the most influential factors on rainfall

partitioning variables at inter-event and intra-event scales, and the corresponding quantitative

relationships were established based on a qualifying level of significance ($p < 0.05$) and the

highest coefficient of determination ($R^2$). Significance levels were set at 95% confidence

intervals. Data analysis was performed using SPSS 21.0, Origin 2018, and Microsoft Excel

2019.

 **3 Results**

**3.1 Inter-event variations of rainfall partitioning**

**3.1.1 Characteristics of inter-event rainfall partitioning variables**

A total of 38 rainfall events were recorded for rainfall partitioning measurements, including

20 events (215.4 mm) in 2014 and 18 events (205.6 mm) in 2015, which accounted for 75.2%

300 and 75.0% of total rainfall amount during the experimental period in 2014 and 2015,

respectively (Fig. 2a). The RA ranged from 1.2 to 41.9 mm with an average of 11.1 ± 8.8 mm

(mean ± standard deviation). In general, rainfall events were unevenly distributed in terms of

RA. Approximately 34.2% of rainfall events were smaller than 5 mm, 26.3% within 5-10 mm,

26.3% within 10-20 mm, and 13.2% larger than 20 mm, representing 8.8%, 17.5%, 36.3%,

305 and 37.4% of the total rainfall amount, respectively (Fig. 2a). The average I varied from 0.2

$mm·h^{-1}$ to 35.1 $mm·h^{-1}$ with an average of 6.0 ± 1.3 $mm·h^{-1}$, and approximately 76.3% of the

events was < 5 $mm·h^{-1}$, 13.2% was 5–10 $mm·h^{-1}$, and 10.5% was > 10 $mm·h^{-1}$. $I_{10\_max}$

ranged from 1.2 $mm·h^{-1}$ to 68.4 $mm·h^{-1}$ with an average of 13.7 ± 2.7 $mm·h^{-1}$, and

approximately 42.1% of the events was < 5 $mm·h^{-1}$, 23.7% was 5–10 $mm·h^{-1}$, and 34.2%

310 was > 10 $mm·h^{-1}$. The RD ranged from 0.2 h to 28.9 h and averaged 5.3 ± 1.0 h. The RD of

most rainfall events was less than 5 h (68.4%), and only 5 rainfall events had RD greater than

10 h.

The $TF_d$ for *C. korshinskii* ranged from 0.7 mm to 31.2 mm (coefficient of variation, CV

= 87.5%) with corresponding TF% ranging from 54.0 to 80.3% (CV = 10.6%) across the 38

events (Fig. 2b). The $TF_d$ values for *S. psammophila* were 0.4-33.4 mm (CV = 96%) and

28.5-82.7% (CV = 21.5%), respectively (Fig. 2c). The $SF_d$ for *C. korshinskii* ranged from 0.04 mm to 6.1 mm (CV = 106.6%), with corresponding SF % of 2.0-14.5% (CV = 34.2%) (Fig. 2b). The comparable $SF_d$ values for *S. psammophila* varied from 0.01 mm to 2.2 mm (CV = 98.6%) and 0.7-5.9% (CV = 38.9%), respectively (Fig. 2c). The $IC_d$ values for *C.*

*korshinskii* varied from 0.5 mm to 2.9 mm (CV = 43.9%), with corresponding IC% of 5.7-40.8% (CV = 47.3%) (Fig. 2b), and the comparable values were 0.8-5.7 mm (CV = 44.8%) and 12.1-70.8% (CV = 53.3%) for *S. psammophila*, respectively (Fig. 2c). For *C. korshinskii*, TF represented the largest component of all rainfall events, while for *S. psammophila*, SF represented the smallest component of all rainfall events (Figs. 2 b and 2c).

The percentages of TF, SF, and IC in rainfall partitioning between two species are shown in Fig. 3. There was no significant difference ($p > 0.05$) in average TF% between *C. korshinskii* (69.4 ± 7.4%) and *S. psammophila* (66.7 ± 14.6%). The SF% was significantly higher ($p < 0.05$) for *C. korshinskii* (9.2 ± 3.2%) than *S. psammophila* (3.8 ± 1.5%) (Fig. 3b). The IC% was significantly lower ($p < 0.05$) for *C. korshinskii* (21.4 ± 10.2%) than *S.*

*psammophila* (29.5 ± 15.9%) (Fig. 3c). The variations of TF% and IC% among the rainfall events were greater for *S. psammophila*, but that of SF% was smaller compared to *C. korshinskii* (Fig. 3).

**3.1.2 Relationships between inter-event rainfall partitioning variables and meteorological factors**

Correlation analysis indicated that meteorological factors had a similar effect on rainfall partitioning for the two species. Stepwise regression analysis identified that the SF

parameters ($SF_d$ and $SF\%$), TF parameters ($TF_d$ and $TF\%$) and IC parameters ($IC_d$ and $IC\%$) were all mainly controlled by RA. Following RA, the influences of rainfall intensity (I, $I_{10\_max}$) were also significant ($p < 0.05$). However, the other meteorological factors (RD, RI, WS, WD, T, H) had no significant effect on rainfall partitioning ($p > 0.05$).

Significantly positive and linear relationships were found between $TF_d$ and RA for both *C. korshinskii* and *S. psammophila* (Fig. 4a). According to the regression equations, the threshold of rainfall amount for TF generation was 0.8 and 1.1 mm for *C. korshinskii* and *S. psammophila*, respectively. The TF% increased with increasing RA as an exponential function (Fig. 4b). When the RA reached 20 mm, the increasing of TF% became stabilized, and TF% of *C. korshinskii* and *S. psammophila* reached 79.2% and 80.0%, respectively. The $SF_d$ also had a significantly positive and linear relationship with RA for the two species (Fig. 4c). When RA was greater than 1.7 mm and 2.2 mm, *C. korshinskii* and *S. psammophila* began to produce SF, respectively. The SF% increased exponentially with increasing RA, and SF% of *C. korshinskii* was always higher than that of *S. psammophila*. The SF% approximately tended to be constant at 12.2% and 5.5% as RA $\geq$ 20 mm for *C. korshinskii* and *S. psammophila*, respectively (Fig. 4d). The $IC_d$ was also positively correlated with RA (Fig. 4e). However, IC% decreased exponentially with incremental RA, and IC% of *S. psammophila* was always higher than that of *C. korshinskii* (Fig. 4f). When RA reached 20 mm, IC% approximately tended to be constant at 9.0% and 14.5% for *C. korshinskii* and *S. psammophila*, respectively.

**3.2 Intra-event variations of rainfall partitioning**

**3.2.1 Characteristics of intra-event rainfall partitioning variables**

The intra-event TF and SF were well synchronized with rainfall process, in terms of the shape,

number and location of the intensity peaks for both *C. korshinskii* and *S. psammophila*, which

was vividly demonstrated at representative four rainfall events in Fig. 5. The SF intensity

($SFI_{10}$) was much higher than TF intensity ($TFI_{10}$) and rainfall intensity ($I_{10}$) for both *C.*

*korshinskii* and *S. psammophila*, whereas $TFI_{10}$ was less than or equal to $I_{10}$. As expected, IC

was the main component at the initial stage of rainfall, and then TF was the major component

($\geq 50\%$) for rainfall partitioning (Fig. 5). The TF and SF generation thresholds measured using

the TBRGs were $0.4 \pm 0.2$ mm and $1.0 \pm 0.7$ mm for *C. korshinskii*, and $0.3 \pm 0.1$ mm and $0.7$

$\pm 0.3$ for *S. psammophila*, respectively. They were expected to be both smaller than the

thresholds derived from the regression equation between $TF_d$ (or $SF_d$) and RA aforementioned

which assume that TF and SF start after the canopy is fully wet. This further demonstrates the

importance of high-resolution data in rainfall partitioning studies.

Fig. 6 describes the difference in average intra-event TF and SF variables between *C*

*korshinskii* and *S. psammophila*. Although there were no statistically significant differences

between the two species in intensities, durations, or the lag time of TF and SF, some trends

were observed. The TFI and $TFI_{10\_max}$ of both species were similar to I ($6.0 \pm 1.3$ mm/h) and

$I_{10\_max}$ ($13.7 \pm 2.7$ mm/h), respectively. In contrast, SFI and $SFI_{10\_max}$ were significantly

greater than I and $I_{10\_max}$, respectively. Specifically, TFI and $TFI_{10\_max}$ of *C. korshinskii* were

$3.8 \pm 1.2$ mm·h$^{-1}$ and $13.3 \pm 4.9$ mm·h$^{-1}$, respectively, which were slightly lower than that of

*S. psammophila* (4.3 ± 1.5 mm·h$^{-1}$ and 14.6 ± 5.5 mm·h$^{-1}$, respectively) (Fig. 6a). The SFI and SFI$_{10\_max}$ of *C. korshinskii* (753.8 ± 208.0 mm·h$^{-1}$ and 3627.2 ± 1424.7 mm·h$^{-1}$, respectively) were higher than those of *S. psammophila* (471.2 ± 170.2 mm·h$^{-1}$ and 1424.8 ± 538.3 mm·h$^{-1}$, respectively) (Fig. 6b).

Furthermore, a time lag was usually observed between the onset of rainfall and the generation of TF (LG$_{TF}$) and SF (LG$_{SF}$). Similarly, there is a time lag between rainfall and TF or SF in terms of the time to reach maximum intensity (LM) and the time to end (LE). The *S. psammophila* had a shorter lag time than *C. korshinskii* in terms of TF (LG$_{TF}$: 0.1 ± 0.04 h vs. 0.3 ± 0.1 h) and SF production (LG$_{SF}$: 0.7 ± 0.3 h vs. 0.8 ± 0.3 h), and their reaching maximum intensity (LM$_{TF}$: 1.8 ± 0.8 h vs. 2.0 ± 0.6 h; LM$_{SF}$: 2.1 ± 0.7 h vs. 2.2 ± 0.5 h) (Figs. 6c and 6d). However, the *S. psammophila* had longer TF duration (5.2 ± 1.4 h vs. 4.8 ± 1.4 h) and LE$_{TF}$ (0.2 ± 0.1 h vs. 0.1 ± 0.1 h) than *C. korshinskii* (Fig. 6c). Conversely, the SF duration and LE$_{SF}$ in *C. korshinskii* (4.6 ± 1.4 h and 0.4 ± 0.1 h, respectively) were longer than those in *S. psammophila* (4.1 ± 1.3 h and 0.2 ± 0.2 h, respectively) (Fig. 6d). The above differences in TF and SF variables indicate that *S. psammophila* should be more conducive to generate TF than *C. korshinskii*, while *C. korshinskii* should be more conducive to produce SF than *S. psammophila*.

**3.2.2 Relationships between intra-event rainfall partitioning variables and meteorological factors**

Similar relationships existed between intra-event rainfall partitioning variables and meteorological factors for two species. For both shrubs, rainfall intensity (I, I$_{10\_max}$, and I$_{10\_b}$)

and RD were the main influencing factors affecting intra-event TF variables (Fig. 7) and SF

variables (Fig. 8). While the effects of other meteorological factors (RD, RI, WS, WD, T, H)

on TF and SF variables within the event were not significant ($p > 0.05$). The TFI, $TFI_{10\_max}$,

$LM_{TF}$, and TFD were linearly correlated with I, $I_{10\_max}$, $LM_R$, and RD, respectively, while

$LG_{TF}$ was power functionally correlated with $I_{10\_b}$ ($p < 0.05$). The TF intensities (TFI and

$TFI_{10\_max}$) of *S. psammophila* increased faster with rainfall intensities (I and $I_{10\_max}$) than that

of *C. korshinskii*. The SFI, $SFI_{10\_max}$, $LM_{SF}$, and SFD were also linearly correlated with I,

$I_{10\_max}$, $LM_R$, and RD, respectively ($p < 0.05$). The $LG_{SF}$ was power functionally correlated

with $I_{10\_b}$ ($p = 0.14$ and $p = 0.16$ for *C. korshinskii* and *S. psammophila*, respectively), which

was weaker than the correlation between $LG_{TF}$ with $I_{10\_b}$. The SF intensities (SFI and

$SFI_{10\_max}$) of *C. korshinskii* increased with rainfall intensities (I and $I_{10\_max}$) more rapidly than

that of *S. psammophila*. However, for both species, there was no significant relationship

between $LE_{TF}$ or $LE_{SF}$ and RD (Figs. 7 and 8). The above results indicate that the intra-event

rainfall partitioning variables largely dependent on rainfall intensity and duration.

## 4    Discussion

### 4.1 Rainfall partitioning and influencing factors at inter-event scale

This study indicated that SF% of *C. korshinskii* (9.2%) was significantly higher than that of *S.*

*psammophila* (3.8%) (Fig. 3), which was comparable to the value of 10.4% and 6.3%

reported by Yang et al. (2019) for the same species in similar semiarid regions of China.

Under the same rainfall regimes, the difference in vegetation characteristics is the main

reason for the difference in SF (Yuan et al., 2017; Whitworth-Hulse et al., 2020a; Yue et al.,

2021). Comparing the structural properties of two shrubs with the same age (20 years), we

found that CA, BD, BL, BA and NB values of *S. psammophila* were 4.51, 1.61, 1.94, 0.83

and 1.52 times of those of *C. korshinskii*, respectively (Table 1). On the branch scale, *C.*

*korshinskii* had more small and short branches, but larger BA than that of *S. psammophila*,

which was contributed to SF generation. Yuan et al. (2016) concluded that a beneficial branch

architecture for SF production should include more relatively small branches and larger

branch angles, and SF productivity decreased with BD size of branches. Furthermore, *C.*

*korshinskii* with a smaller CA, and it had a larger $SF_d$ than *S. psammophila* under the

same SF volume. Somewhat in line with Yuan et al. (2016) and Yue et al. (2021), our

results suggest that a beneficial branch architecture for SF production of *C. korshinskii* should

include relatively small CA, BD, BL and large BA (Table 1).

Leaf traits had been reported to exert a significant influence on rainfall partitioning

(Garcia-Estringana et al., 2010; Magliano et al., 2019a). According to the documentation in

*Flora of China* (Liu et al., 2010), *C. korshinskii* has pinnate compound leaves and each pinna

has 5 to 8 pairs of ovate leaflets, and the leaves are lanceolate and concave, and the surface is

densely sericeous. In comparison, *S. psammophila* has stripe or stripe oblanceolate leaves,

margin revolute, and which upper surface of mature leaf blade is almost glabrous (Chao and

Gong, 1999). The branches of both shrubs are smooth, with a more developed cuticle layer on

the surface of the *S. psammophila* branches, while the *C. korshinskii* branches contain oil and

have waxy skin (Chao and Gong, 1999; Liu et al., 2010). The leaf morphology and epidermal

characteristics of branches of *C. korshinskii* was more beneficial for SF generation than that

of *S. psammophila* (Whitworth-Hulse et al., 2020b; Yuan et al. 2017). It was found that big biomass of leaves, concave leaf shape and leaf pubescence are beneficial to promote the generation of SF (Yuan et al., 2016). These factors together enable the leaves to function as a highly efficient natural water collecting system.

The mean IC% of *C. korshinskii* (21.4%) was significantly lower than that of *S. psammophila* (29.5%) in this study. The intercepts in the fitted formulas between interception loss and rainfall amount in Fig. 4e indicated that *C. korshinskii* (0.92 mm) had a lower canopy water storage than *S. psammophila* (1.15 mm), hence the potential interception loss of *C. korshinskii* was lower. Zhang et al. (2017) reported that IC% were higher in the *H.*

*rhamnoides* stand (24.9%) than in the *S. pubescens* stand (19.2%), which was mainly attributed to the lower canopy water storage of *S. pubescen*. This study was done at the shrub-scale, so we compared the total canopy water storage of individual plant ($C_m$), and we found that $C_m$ of *S. psammophila* (6.0 L) was significantly higher than that of *C. korshinskii* (3.9 L) (Table 1). This was mainly due to the significantly higher average total dry

aboveground biomass of *S. psammophila* (15.7 kg per plant) than *C. korshinskii* (4.6 kg per plant). Consequently, individual *S. psammophila* absorbed more rain water to moisten the branches and leaves than that of individual *C. korshinskii*, which could explain higher IC% of *S. psammophila* than *C. korshinskii*. Thus, the best predictors for interception loss were biomass-related parameters (i.e., woody biomass and total biomass) (Li et al., 2016).

**4.2 Rainfall partitioning and influencing factors at intra-event scale**

Temporal heterogeneity of rainfall clearly influences the amount and timing of TF and SF

reaching the soil under the canopy, as explained by some previous intra-event rainfall partitioning studies from forested ecosystems (Owens et al., 2006; Levia et al., 2010; Molina et al., 2019). Our experiment investigated the intra-event dynamics of all rainfall partitioning

components in xerophytic shrubs, which has scarily been reported before. Our results showed that the temporal dynamics of TF and SF under the shrub canopy almost matched the dynamics of rainfall (Fig. 5). It agreed with the reports of Zhang et al. (2018) and Yuan et al. (2019) who demonstrated the temporal synchronization of TF and SF with rainfall, respectively. The SF intensity is generally greater than rainfall intensity for different species

(Fig. 6), which can influence ecohydrological processes such as groundwater recharge, erosion, and overland flow (Spencer and van Meerveld, 2016). The SF converges substantial rainwater to the shrub bases and then delivers it into the soil as a point input to recharge soil moisture and nutrient enrichment (Germer et al., 2010; Wang et al., 2011; Cayuela et al., 2018; Jian et al., 2019). Moreover, as stemflow water is funneled belowground along roots of the

shrubs (Martinez-Meza and Whitford, 1996; Li et al., 2009), we suggest that changes in SF inputs explain, at least in part, the temporal variation in subsurface moisture patterns.

The intensity variables and lag time of SF and TF relative to rainfall were the key to describe the intra-event rainfall partitioning (Fig. 6). The effects of meteorological factors on SF and TF variables at the intra-event scale were derived from multiple regression analysis in

this study. The SF and TF variables (intensity and temporal dynamics) were strongly influenced by rainfall intensity (e.g., I, $I_{10\_max}$ and $I_{10\_b}$) and duration (e.g., RD and $LM_R$). This is consistent with the results reported by Yuan et al. (2019) who indicated that there was

a significant effect of rainfall intensity on the stemflow process of *C. korshinskii*. The main factors affecting intra-event SF and TF variables were the same, but the effects were still slightly different between the two shrubs. Under the same rainfall intensity, the average TF intensity under the canopy of *S. psammophila* was higher than *C. korshinskii* (Figs. 7a and 7b). But the average SF intensity of *C. korshinskii* was greater than *S. psammophila* at shrub scale (Figs. 8a and 8b), which was also found for the branch SF intensity reported by Yuan et al. (2019). In addition to the inter-shrub differences, the effects of $I_{10\_b}$ on $LG_{TF}$ and $LG_{SF}$ were slightly different. The correlation between $LG_{SF}$ and $I_{10\_b}$ (Fig. 8c) was weaker than that between $LG_{TF}$ and $I_{10\_b}$ (Fig. 7c). This may be due to the fact that TF has two components, i.e., free TF and released TF (Staelens et al., 2008; Levia et al., 2017; Van Stan II et al., 2020), and that SF only starts to produce when a certain amount of rainfall is reached (Germer et al., 2010; Levia et al., 2010; Dunkerley, 2014; Yuan et al., 2019). Our results indicated that *S. psammophila* had dynamic characteristics (e.g., larger TFI, $TFI_{10}$ and $LE_{TF}$ as well as TFD, and shorter $LG_{TF}$ and $LM_{TF}$) producing larger TF depth (TFd = TFI*TFD) (Figs. 6a and 6c), while *C. korshinskii* had dynamic characteristics (e.g., larger SFI, $SFI_{10}$ and $LE_{SF}$ as well as SFD) producing larger SF depth (SFd = SFI*SFD) (Figs. 6b and 6d).

The vegetation characteristics have an important effect on the dynamics and the lag time of TF and SF (Yuan et al., 2019; Zhang et al., 2018). Based on the temporal data recorded by TBRGs, we found that *C. korshinskii* produced TF and SF later than *S. psammophila* (Figs. 5 and 6), which was also reported by Yuan et al. (2019) for branch SF of the same species. We inferred that this was due to the higher canopy water storage capacity of *C. korshinskii* (0.85

mL/g) compared to *S. psammophila* (0.38 mL/g). However, when the branches were

moistened, SF production of *C. korshinskii* was greater than that of *S. psammophila* because

of its branch and leaf characteristics as discussed in subsection 4.1 (Fig. 5). It was found that

the great bark water storage capacity of forests could result in the further delay of TF and SF

onset (Levia and Herwitz, 2005; Levia et al., 2010; Li et al., 2016; Pinos et al., 2021). In

summary, the different intra-event TF and SF dynamics between species were attributed to a

complex interaction of biotic and abiotic factors (Yuan et al., 2019; Zhang et al., 2018; Levia

et al., 2010).

## 4.3 Implications and further scopes

Most of previous rainfall partitioning investigations for shrubs were limited at inter-event scale,

or only focused on TF or SF at intra-event scale. The intra-event rainfall partitioning dynamics,

which could help have a better understanding of soil water replenishment and its distribution in

soil and the key ecohydrological cycle in in drylands, have been rarely explored. This study is

the first time to investigate all the rainfall partitioning components (i.e., TF, SF and IC) for

shrubs at both inter- and intra-event scales, which steps further and provides a full view of the

reciprocal dynamics among interception loss, throughfall, and stemflow at the shrub-scale.

This is the main novelty and a step forward compared with the previous related studies. We

have also obtained the quantitative relationship between rainfall partitioning variables and

rainfall characteristics, and further elaborated the influence of vegetation structure

characteristics (leaf, canopy structure, and biomass, etc..) on rainfall partitioning. The obtained

new insights help to understand the fine characterization of shrub-dominated eco- hydrological

processes, and improve the accuracy of water balance estimation in dryland ecosystem.

There are several issues that need further investigation. Firstly, long-term observations of rainfall partitioning dynamics for more shrub plants and species are needed, and the rainfall partitioning models should be developed for shrubs. Every component of shrub canopy water balance including canopy evaporation loss and transpiration should be considered. Secondly,

the effects of rainfall partitioning on soil moisture dynamics, nutrient cycling, and plant transpiration should be substantially investigated. How the shrubs actually make use of small amounts of throughfall or stemflow should be examined to detect the interactions between water redistribution and vegetation physiological processes. Finally, the extension from the individual plant to stand and larger scales remains a challenging topic for rainfall partitioning,

which can help improve understanding the role of rainfall partitioning in the regional hydrologic cycle.

## 5 Conclusions

In this study, we analyze the rainfall partitioning and the influences of bio-/abiotic factors of two typical shrubs at both inter- and intra-event scales in the Loess Plateau. To ensure a larger

proportion of the rainfall is allocated under the canopy, two species can obtain more net rainfall through different mechanisms. At the event scale, there was no significant difference in TF percentage between the two shrubs, but *C. korshinskii* had significantly higher SF percentage and lower IC percentage compared to *S. psammophila*. At the intra-event scale, TF and SF of two shrubs were well synchronized with the rainfall, but *C. korshinskii* had the advantage of

stemflow production, while *S. psammophila* had the advantage of TF generation. For both

shrubs, the inter-event rainfall partitioning amount and percentage depended more on rainfall amount, and rainfall intensity and duration controlled the intra-event TF and SF variables. The *C. korshinskii* has larger branch angles, more small branches and smaller canopy areas to produce SF more efficiently, and *S. psammophila* has larger biomass to intercept more rainfall

amount. These findings could enhance our understanding of TF and SF dynamics and corresponding driving factors at inter- and intra-event scales, and help in modelling the critical eco-hydrological processes in arid and semi-arid regions.

**Data availability:** The data that support the findings of this study are available from the

corresponding author upon request.

**Author contributions:** JXA: Formal analysis, Investigation, Methodology, Writing - original draft; GYG: Conceptualization, Methodology, Writing - review & editing; CY: Investigation, Writing - review & editing; JP and BJF: Writing - review & editing.


**Competing interests:** The authors declare that they have no conflict of interest.

**Acknowledgements:** This research was supported by the National Natural Science Foundation of China (nos. 41991233 and 41822103), and the Youth Innovation Promotion Association CAS (no. Y202013). Special thanks are given to Shenmu Erosion and Environment Research Station for experimental support to this research. We thank David Dunkerley and two anonymous reviewers for their professional comments which greatly improved the quality of this paper.

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

**Table 1.** Descriptive statistics (mean ± standard error) of canopy morphology of *C. korshinskii* (CK1-CK3) and *S. psammophila* (SP1-SP3) plants. Values are mean ± SD.

| Plant ID | SH (m) | CA (m$^2$) | NB | BL (cm) | BA (°) | BD (mm) | TBA (cm$^2$) | TB (kg) | $C_m$ (L) |
|---|---|---|---|---|---|---|---|---|---|
| CK1 | 2.2 | 5.3 | 47 | 150.6±5.1 | 60.26±2.6 | 9.2±0.5 | 34.9 | 4.0 | 3.4 |
| CK2 | 2.3 | 5.2 | 47 | 123.3±6.6 | 65.5±2.1 | 8.5±0.5 | 31.3 | 3.6 | 3.1 |
| CK3 | 2.4 | 5.3 | 49 | 134.6±6.7 | 65.4±4.4 | 9.9±0.7 | 45.8 | 6.2 | 5.2 |
| Average | 2.3a | 5.27a | 48a | 136.2a | 63.77a | 9.2a | 37.3a | 4.6a | 3.9a |
| SP1 | 3.5 | 23.9 | 85 | 262.2±6.0 | 67.1±1.4 | 13.8±0.5 | 139.7 | 14.3 | 5.4 |
| SP2 | 3.3 | 26.1 | 55 | 268.0±7.7 | 56.0±3.1 | 15.1±0.6 | 124.3 | 15.9 | 6.0 |
| SP3 | 3.6 | 21.4 | 78 | 262.0±7.8 | 35.1±2.9 | 15.3±0.5 | 155.9 | 17.0 | 6.5 |
| Average | 3.5b | 23.8b | 73b | 264.1b | 52.7a | 14.8b | 140.0b | 15.7b | 6.0b |

Note: SH: shrub height; CA: canopy area; NB: number of branches; BL: branch length; BA: branch angle; BD: basal diameter of branch; TBA: total basal area of the shrub; TB: total dry aboveground biomass; $C_m$: total canopy storage per plant. Different letters indicate statistically significant differences between two

species ($p < 0.05$).

**Table 2.** Rainfall partitioning parameters at inter- and intra-event scales.

| Scale | Parameter (unit) | Explanation |
|---|---|---|
| Inter-event | $TF_d$ (mm) | Throughfall depth per rainfall event |
| | $SF_d$ (mm) | Stemflow depth per rainfall event |
| | $IC_d$ (mm) | Interception loss depth per rainfall event |
| | TF% | Percentage of TF per rainfall event |
| | SF% | Percentage of SF per rainfall event |
| | IC% | Percentage of IC per rainfall event |
| | TFD (h) | Throughfall duration |
| | SFD (h) | Stemflow duration |
| | TFI (mm·h$^{-1}$) | Average throughfall intensity |
| | SFI (mm·h$^{-1}$) | Average stemflow intensity |
| Intra-event | $I_{10}$ (mm·h$^{-1}$) | Rainfall intensity at 10-min interval |
| | $I_{10\_max}$ (mm·h$^{-1}$) | Maximum $I_{10}$ during the rainfall process |
| | $TFI_{10}$ (mm·h$^{-1}$) | Throughfall intensity at 10-min interval |
| | $TFI_{10\_max}$ (mm·h$^{-1}$) | Maximum $TFI_{10}$ during the rainfall process |
| | $SFI_{10}$ (mm·h$^{-1}$) | Stemflow intensity at 10-min interval |
| | $SFI_{10\_max}$ (mm·h$^{-1}$) | Maximum $SFI_{10}$ during the rainfall process |
| | $LG_{TF}$ (h) | Time lag of throughfall generation after the start of rainfall |
| | $LG_{SF}$ (h) | Time lag of stemflow generation after the start of rainfall |
| | $LM_R$ (h) | Time lag of $I_{10\_max}$ occurrence relative to the onset of rainfall |
| | $LM_{TF}$ (h) | Time lag of $TFI_{10\_max}$ occurrence relative to the onset of rainfall |
| | $LM_{SF}$ (h) | Time lag of $SFI_{10\_max}$ occurrence relative to the onset of rainfall |
| | $LE_{SF}$ (h) | Time lag of throughfall ending after the end of rainfall |
| | $LE_{SF}$ (h) | Time lag of stemflow ending after the end of rainfall |

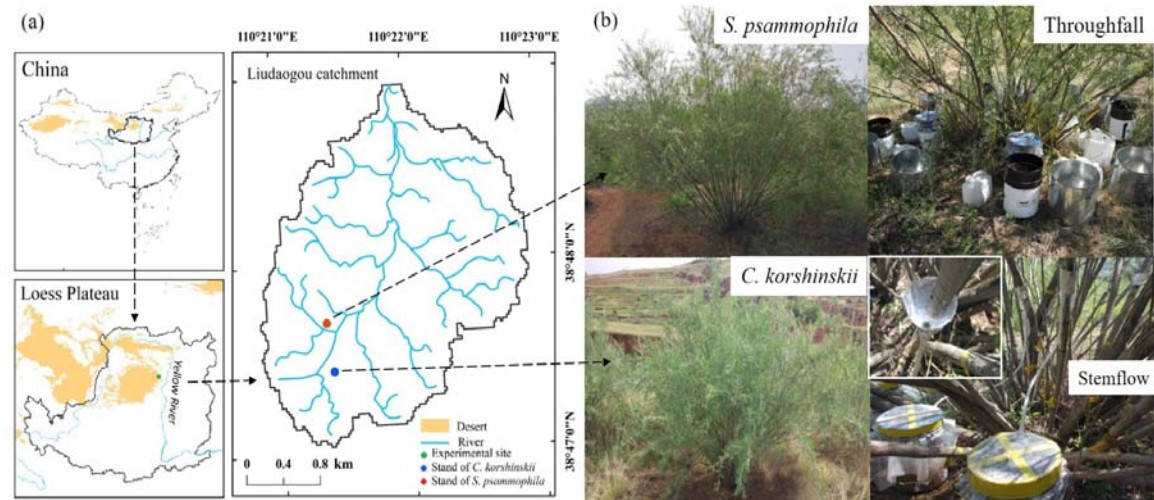

**Figure 1.** The location and experimental settings in the plots of *C. korshinskii* and *S.*

*psammophila.*

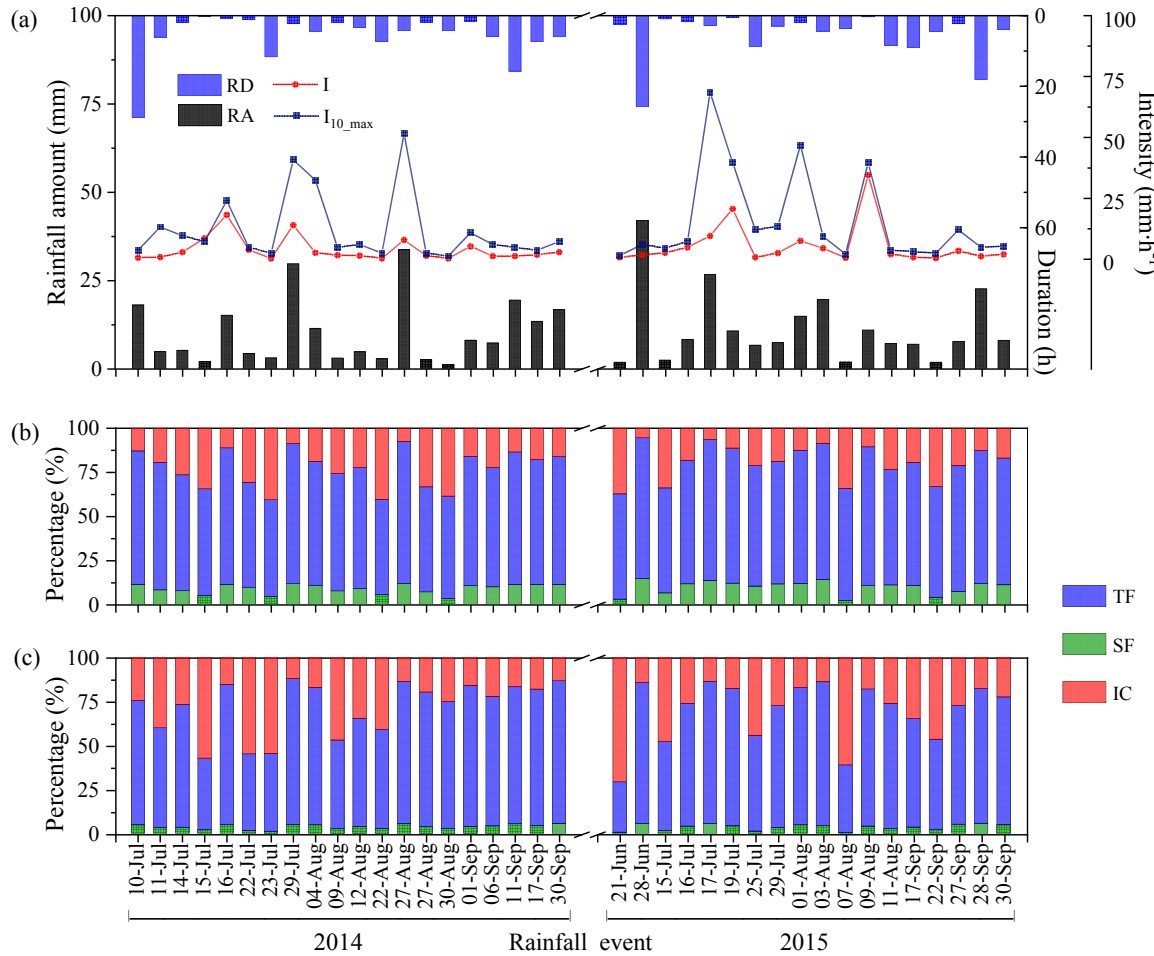

**Figure 2**. (a) individual rainfall amount (RA) (*n* = 38), rainfall duration (RD), average rainfall intensity (I, mm·h$^{-1}$), maximum rainfall intensity at 10-min interval (I$_{10\_max}$, mm·h$^{-1}$); and rainfall partitioning into TF %, SF %, and IC % of (b) *C. korshinskii* and (c) *S. psammophila*.

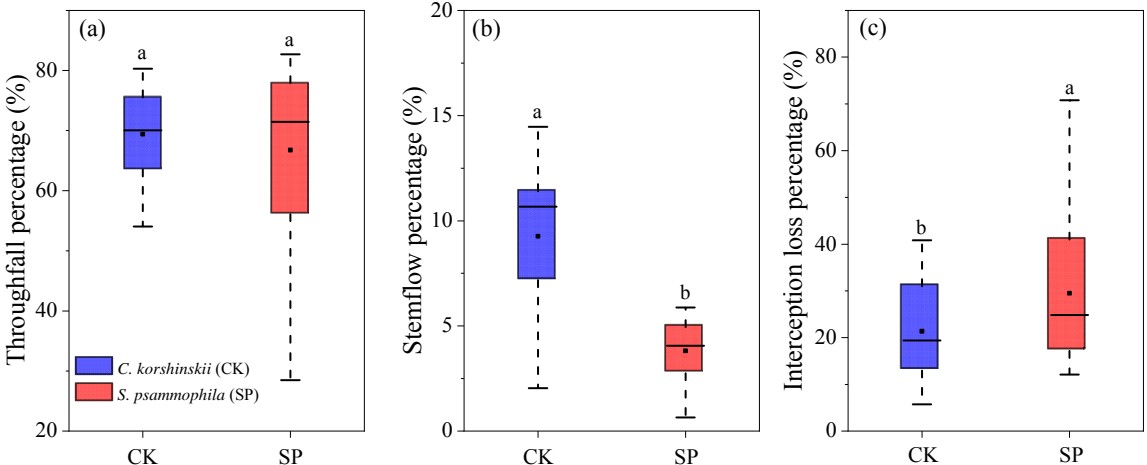

**Figure 3**. Box-plots of (a) TF%, (b) SF%, and (c) IC% for *C. korshinskii* (CK) and *S. psammophila* (SP). The horizontal thick black line indicates the median, boxes correspond to the 25th and 75th percentiles, and whiskers represent values that fall within 1.5 times the interquartile range. Mean values are represented with the black square. Different letters indicate significant differences between the two species ($p < 0.05$).

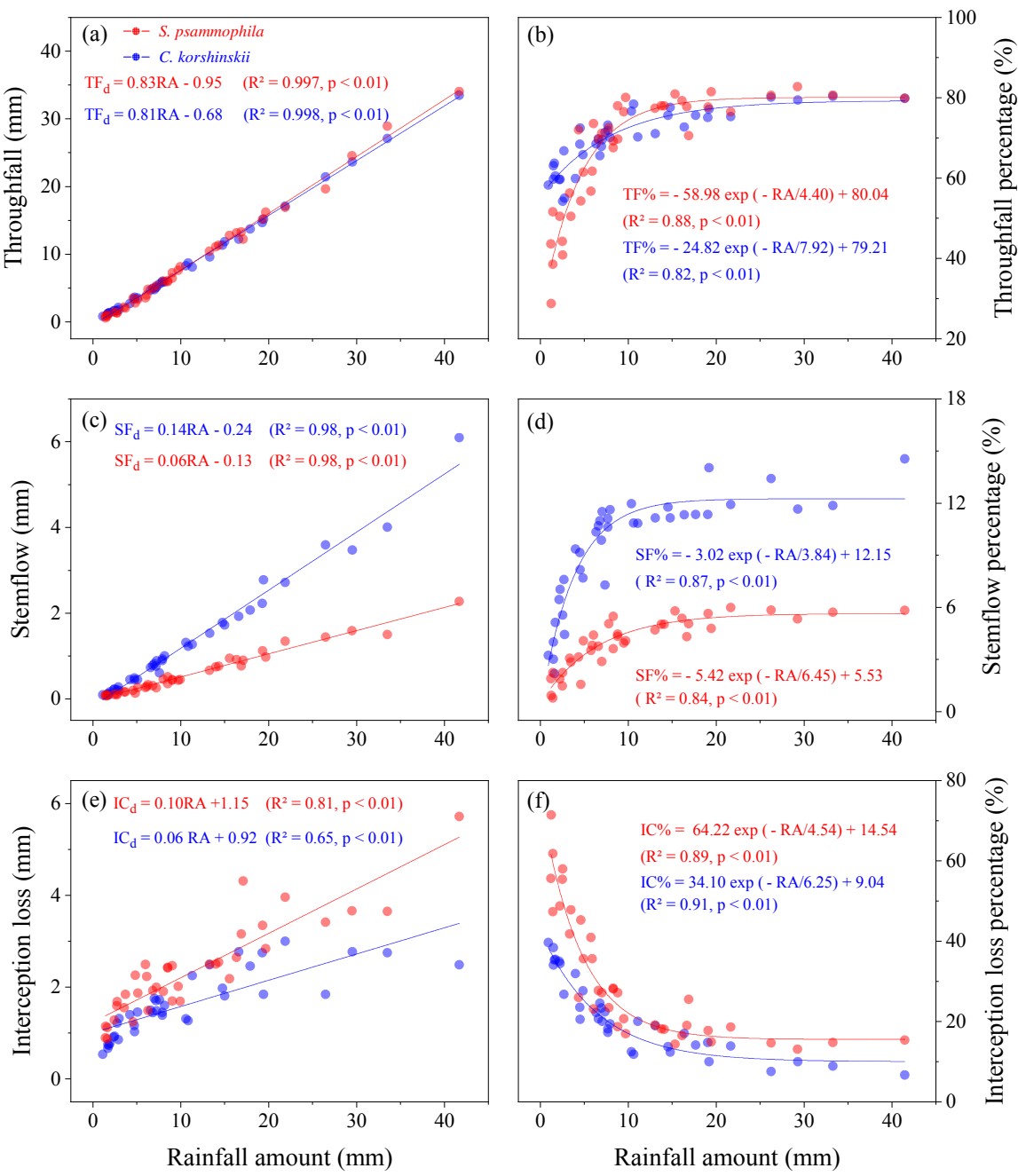


**Figure 4.** Inter-event rainfall partitioning as a function of individual rainfall amount for *C. korshinskii* and *S. psammophila*.

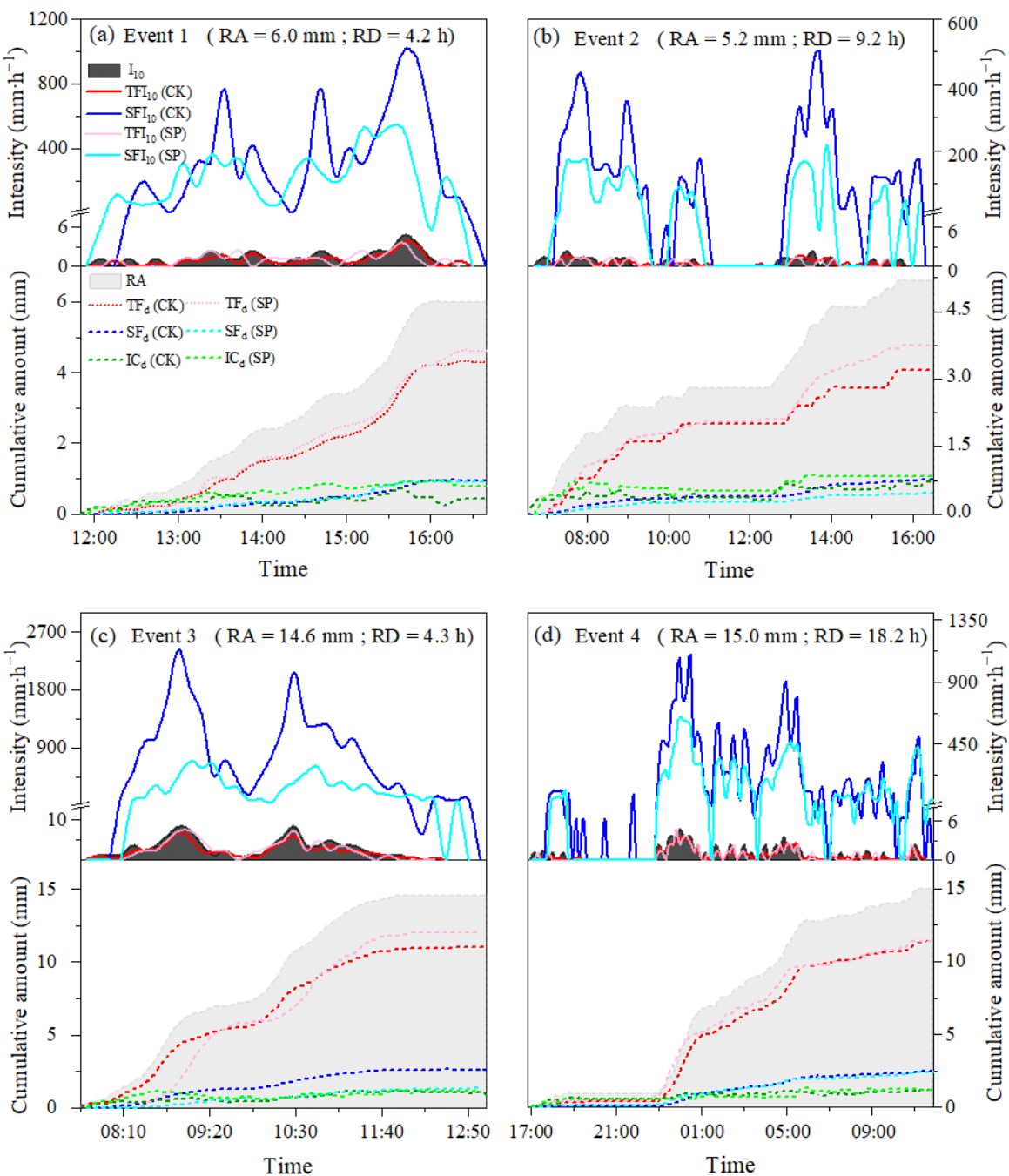

**Figure 5.** Time series (10-min interval) of rainfall partitioning within four rainfall events for *C. korshinskii* (CK) and *S. psammophila* (SP). Events 1-4 occurred on August 3, September 17, September 28, and September 30 in 2015, respectively. The solid lines represent the rainfall, TF and SF intensity at 10-min interval. The dotted lines indicate the accumulated amount of RA, TF, SF, and IC.

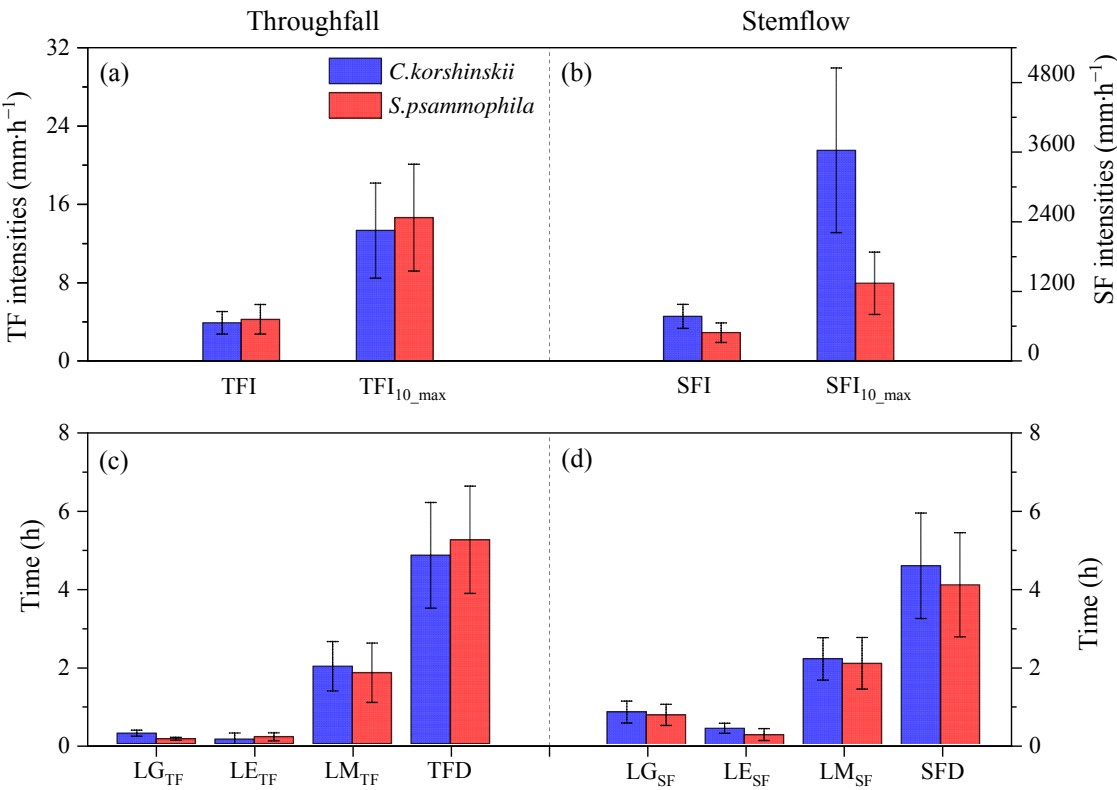


**Figure 6.** Intra-event TF (a, c) and SF (b, d) variables of *C. korshinskii* and *S. psammophila.*

All the variables are explained in Table 2.

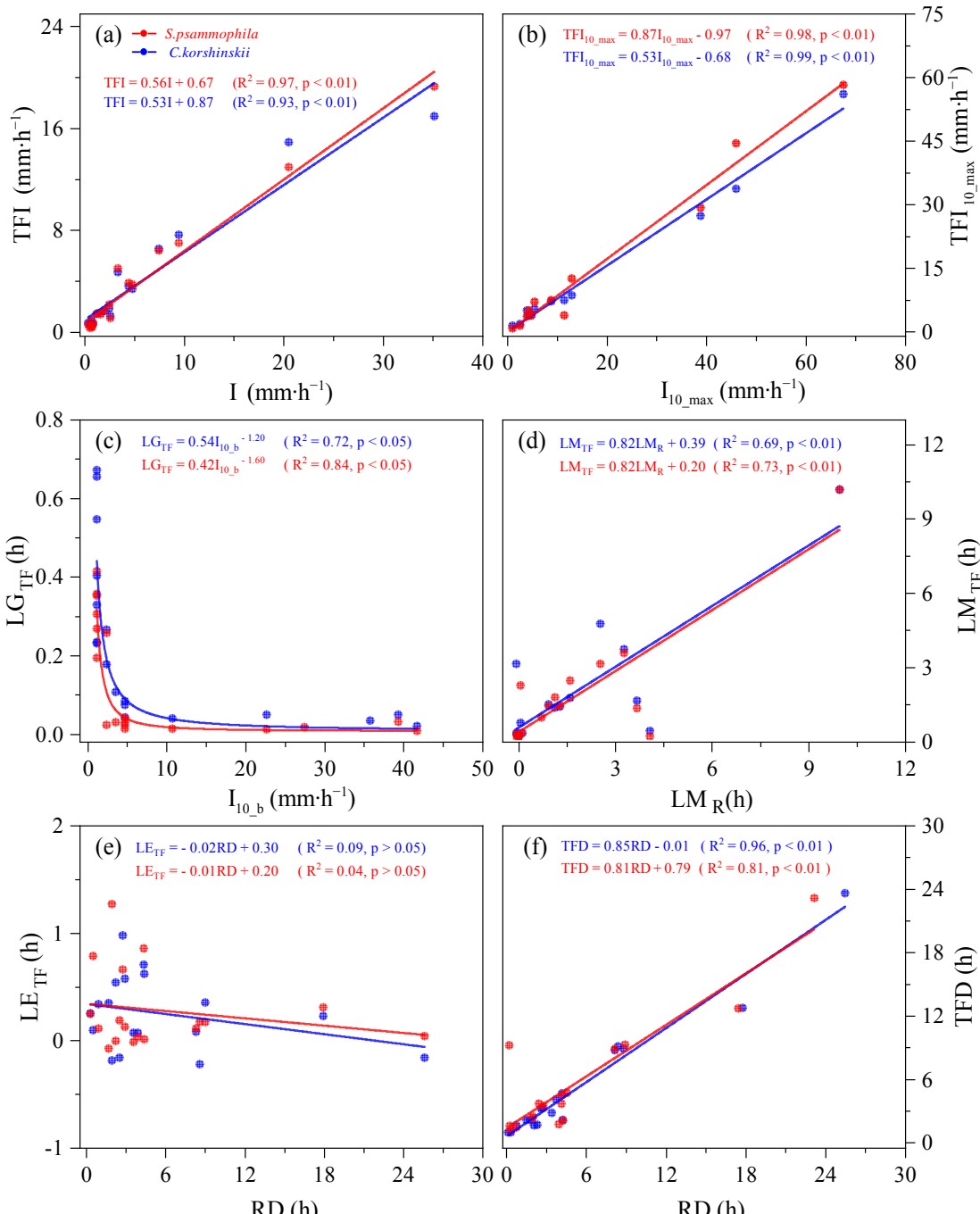

**Figure 7.** Relationships of intra-event throughfall variables with meteorological characteristics for *C. korshinskii* and *S. psammophila*. All the variables are explained in Table 2.

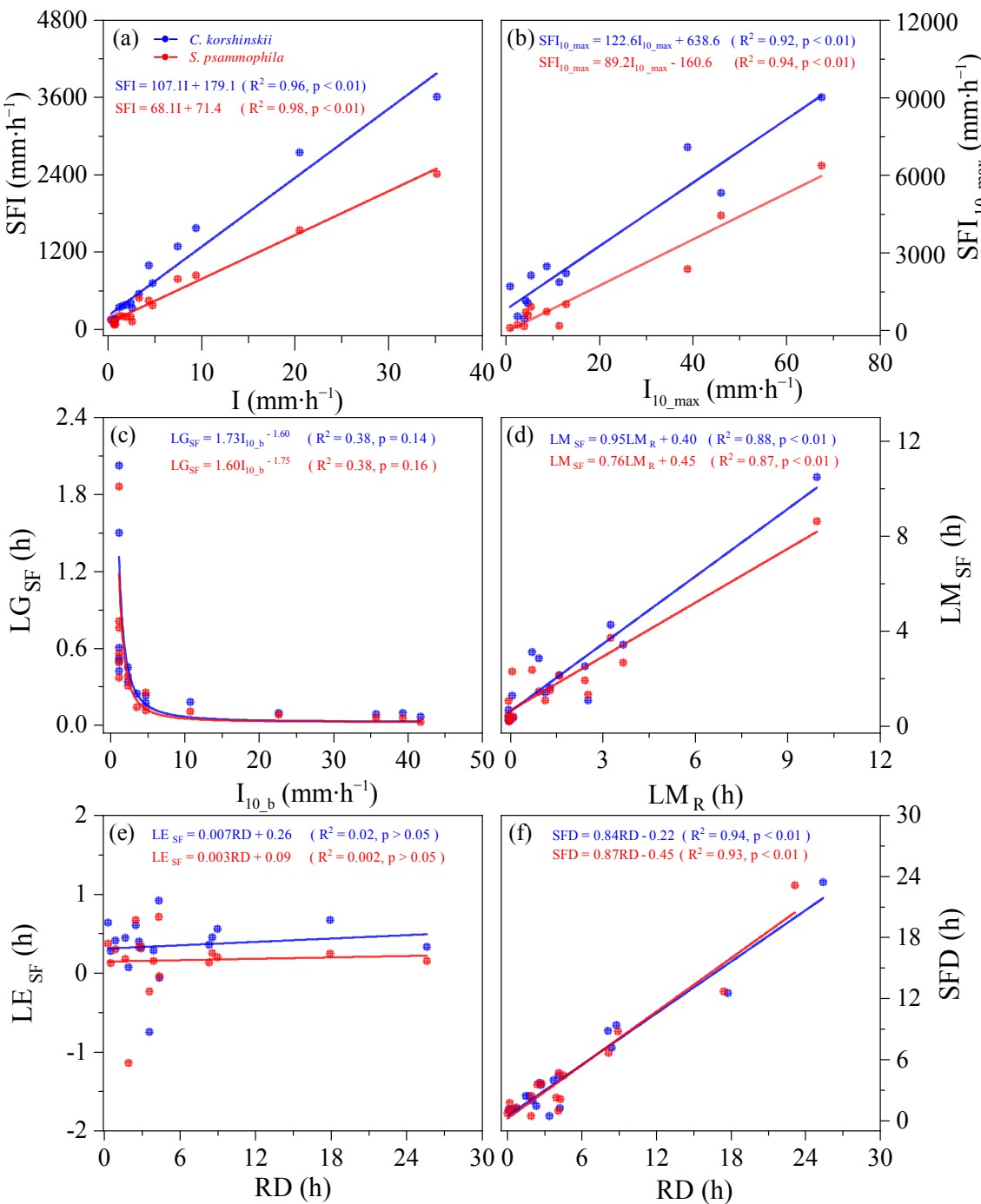

**Figure 8**. Relationships of intra-event stemflow variables with meteorological characteristics for *C. korshinskii* and *S. psammophila*. All the variables are explained in Table 2.