# Peer review of "Inter- and intra-event rainfall partitioning dynamics of two typical xerophytic shrubs in the Loess Plateau of China"

_Hydrology and Earth System Sciences, 2022_

## Author Comment (AC2)

We have carefully addressed all the comments made by David Dunkerley and the two anonymous reviewers on our manuscript (hess-2022-4) entitled "Inter- and intra-event rainfall partitioning dynamics of two typical xerophytic shrubs in the Loess Plateau of China". The comments have helped us greatly improve the overall quality of the manuscript. The following is the point-point response to all the comments.

**Response to Referee #1 (Pro. David Dunkerley):**

**1. Comment:**

This paper explores a little further data from the same field observations as were discussed previously by some of the same authors in their 2019 HESS paper *Temporally dependent effects of rainfall characteristics on inter- and intra-event branch-scale stemflow variability in two xerophytic shrubs* (Yuan et al. 2019).

[Yuan, C., Gao, G., Fu, B., He, D., Duan, X., & Wei, X. (2019). *Temporally dependent effects of rainfall characteristics on inter- and intra-event branch-scale stemflow variability in two xerophytic shrubs. Hydrol. Earth Syst. Sci.*, 23(10), 4077-4095. doi:10.5194/hess-23-4077-2019].

The same two species of shrubs are studied, and the field observations analysed here come from the same 2014-2015 data collection as were analysed by Yuan et al. (2019). The field data collection appears to have been one and the same for both papers.

Reply: We acknowledge that this work and Yuan et al. (2019) did analyze the same two species of shrubs in the same area during 2014-2015, but our dataset was more complete, including data collection on throughfall and stemflow, as well as interception loss based on water balance. Yuan et al. (2019) only studied the branch-scale stemflow (only one of the rainfall partitioning processes), which can be seen from the title of Yuan et al. (2019). The research object of this work was individual shrub rather than branches, and the aim was to investigate in-deep the holistic processes of rainfall partitioning (i.e., throughfall, stemflow, and interception loss) at intra-event scales, for two typical xerophytic shrubs and to add some new insights into rainfall partitioning. To our best knowledge, this study is the first time to investigate the intra-event variations of the whole rainfall partitioning components for shrubs, on a single plant scale. This is the major novelty and advancement of this study. We have obtained the quantitative relationship between the rainfall partitioning variables and rainfall characteristics, and further elaborated the influences of vegetation structure characteristics (leaf, canopy structure, and biomass, etc..) on rainfall partitioning in the Discussion section.

**2. Comment:**

Both Yuan et al. (2019) and the present ms. (An et al.) seek to explore the role of rainfall variability and plant architecture on stemflow, throughfall, and interception, paying attention to how these work at intra-event timescales. Their ability to do this is however hampered by their reliance on rainfall observations that were aggregated and logged only every 10 minutes. This is hardly sufficient temporal resolution to permit analysis of time lags before the commencement of stemflow, and various other analyses that the authors seek to make.

Reply: Yuan et al. (2019) only explored the branch stemflow variations at intra-event timescale, and this study investigate the intra-event variations of the whole rainfall partitioning components on the plant scale (including throughfall, stemflow, and interception loss).

In terms of temporal resolution, the rainfall partitioning data at intra-event scale was recorded every 0.2 mm using tipping bucket rain gauge (TBRG). The data of stemflow and throughfall volume and timing were automatically recorded at dynamic intervals between neighboring tips. The observations data were aggregated and recorded every 10 minutes to better reflect fluctuations in rainfall partitioning components. A series of indices reflecting the dynamic process of rainfall partitioning were obtained.

**3. Comment:**

Given that the two papers explore the same shrub taxa in the same field area during the same two years (2014-2015), and that both explore the intra-event workings of stemflow and throughfall, I think that a key requirement is for the Introduction to make it clear and explicit how the present paper differs in scope and results from the earlier paper of Yuan et al. (2019). The earlier paper appears to have focussed more strongly on branch-scale mechanisms, but a clear distinction requires very careful reading and differences in data processing make it very difficult indeed to see what is new in the current ms.

Reply: Yuan et al. (2019) only explored the branch stemflow variations at intra-event timescale, and this study investigate the intra-event variations of the whole rainfall partitioning components on the plant scale (including throughfall, stemflow, and interception loss). In the Introduction section, we have made clear the differences between the present paper and the earlier paper by Yuan et al. (2019). The innovation of this study was to investigate in-deep the holistic processes of whole rainfall partitioning components (throughfall, stemflow, and interception loss) of shrub plant at inter- and intra-event scales.

**4. Comment:**

In particular, in their Introduction (and perhaps also in a covering letter to accompany their submission) the authors should highlight what can be learned about stemflow and throughfall in the two shrub taxa that was not already demonstrated by Yuan et al. 2019). I think that it would be helpful for the authors to compare and contrast what was learned by Yuan et al. (2019) and what similarities or differences emerge in the present study (An et al.).

Reply: This is a good comment. Yuan et al. (2019) described the stemflow variability at the branch scale and quantified its relationship with rainfall characteristics on inter- and intra-event. However, they did not study the variability of throughfall (the largest component of rainfall partitioning) and ignored interception loss. This work further provided the dynamics of the whole rainfall partitioning including stemflow, throughfall, and interception loss at the individual plant scale. In the Introduction section, we highlight the differences between this study and Yuan et al. (2019), and discuss in detail the innovative and important implications of this study in the Discussion section. For the results of stemflow, the two papers were largely similar, both in describing stemflow processes and in quantifying their relationship to the rainfall characteristics within event.

**5. Comment:**

In other respects the paper seems entirely routine, containing nothing new in method, theory, or argument. I do think that the authors should evaluate the adequacy f their data and field sampling, however. Do the 38 rainfall events in two years form a sufficiently large sample of events? Does sampling of just four branches (line 202) represent a sufficiently large sample? What is the evidence for this being the case? And only a single individual plant of each species was used to measure throughfall and stemflow dynamics (line 198). How was the single individual selected? Is a sample of one really sufficient to draw meaningful conclusions concerning the entire species, as the authors do? These matters and similar considerations should be discussed and critically evaluated. As the paper stands, the authors merely report that they studied only a single shrub of each species but provide no evaluation of whether this is a sufficient and representative sample. (In the same vein, the use of aggregated, 10-minute rainfall amounts warrants critical comment by the authors. The authors should also report fully and properly the characteristics of the rainfall events that they monitored. To judge from the data in Yuan et al. (2019) these were mostly rather brief - from one to a few hours. But the present ms. (An et al.) does not even mention the event duration (nor, for instance, whether the rain was during daylight hours or at night - which is surely relevant to evaporative losses and hence to interception amounts). All of this must be corrected in a revision to the current ms. Some evidence of the nature of just four rainfall events can be found in Figure 5 but this is hardly sufficient.

Reply: First, this is the first comprehensive study of intra-event dynamics of rainfall partitioning components (i.e., throughfall, stemflow and interception loss) in shrubs. This is the main novelty and advance of this study. We have obtained the quantitative relationship between rainfall partitioning variables and rainfall characteristics, and further elaborated the influence of vegetation structure characteristics (leaf, canopy structure, and biomass, etc..) on rainfall partitioning.

Second, in terms of the number of rainfall events and sampling methods, we refer to studies by Magliano et al., 2019b, Whitworth-Hulse et al., 2020a, and Yang et al., 2019, which studied 22-40 rainfall events. So that we think 38 rainfall events in two years constitute a relatively sufficiently large event sample. And we added the characteristics of the rainfall events in the paper. The start and end time and duration of rainfall were added in Fig. 5 (see Fig. 5). We also pointed out the need for long-term observational studies in the Discussion section.

Third, we measured branch diameter (BD, mm), length (BL, cm), and angle (BA, o) of all 143 and 218 branches of C. korshinskii and S. psammophila. And the BD categories are defined as 0-10, 10-15, 15-20, and >20mm to ensure an appropriate number of branches within the category. On this basis, for each species, we selected six representative branches to characterize intra-event stemflow dynamics through strict selection conditions, including those branches be distributed across the four BD categories and that there was no crossover between the experimental branch and adjacent branches, no inflection point from the tip of the branch to the base, and easy to measure. Due to missing or incomplete data, four branches were finally identified for each species, located in each of the four BD categories to measure stemflow. Branch information is supplemented in the article. The single individual shrub was selected from the three representative shrub plants with similar crown heights and crown areas in each shrub species. It was selected to measure the dynamics of intra-event rainfall partitioning, mainly because of the high cost of equipment, and the difficulty of placing a lot of TBRGs. We compared the mean depth of throughfall and stemflow measured by TBRGs with the mean measured manually and found no significant difference between the two methods. In the future experiment, we will add more branches and individual shrubs to expand the sample size.

Finally, we have explained that rainfall observations were summarized and recorded every 10 minutes to better reflect fluctuations in rainfall partitioning rather than the accuracy of our data recording. The accuracy of our data logging is recorded every 0.2 mm.

| Species and    |    | The generation thresholds (mm) |             | Average depth (mm) |             |
|----------------|----|--------------------------------|-------------|--------------------|-------------|
|                |    | Inter-event                    | Intra-event | Inter-event        | Intra-event |
| C. korshinskii | TF | 0.8                            | $0.4\pm0.2$ | $8.1\pm7.9$        | $7.2\pm5.4$ |
|                | SF | 1.7                            | $1.0\pm0.7$ | $1.2 \pm 1.3$      | $1.3\pm1.1$ |
| S. psammophila | TF | 1.1                            | $0.3\pm0.1$ | $8.3\pm7.4$        | $7.5\pm5.7$ |
|                | SF | 2.2                            | $0.7\pm0.3$ | $0.5\pm0.5$        | $0.9\pm0.8$ |

Table 3. The generation thresholds and mean depth of rainfall partitioning by *C. korshinskii* and *S. psammophila* at inter- and intra-event scales. Values are mean  $\pm$  SD.

At inter-event scale, the thresholds for TF and SF generation are derived from the regression equation in Fig. 4. At intra-event scale the threshold values are measured using tipping bucket rain gauges.

---

## Author Comment (AC4)

We have carefully addressed all the comments made by David Dunkerley and the two anonymous reviewers on our manuscript (hess-2022-4) entitled "Inter- and intra-event rainfall partitioning dynamics of two typical xerophytic shrubs in the Loess Plateau of China". The comments have helped us greatly improve the overall quality of the manuscript. The following is the point-point response to all the comments.

Response to Referee #1 (Pro. David Dunkerley):

1. Comment:

This paper explores a little further data from the same field observations as were discussed previously by some of the same authors in their 2019 HESS paper *Temporally dependent effects of rainfall characteristics on inter- and intra-event branch-scale stemflow variability in two xerophytic shrubs* (Yuan et al. 2019).

[Yuan, C., Gao, G., Fu, B., He, D., Duan, X., & Wei, X. (2019). *Temporally dependent effects of rainfall characteristics on inter- and intra-event branch-scale stemflow variability in two xerophytic shrubs. Hydrol. Earth Syst. Sci.*, 23(10), 4077-4095. doi:10.5194/hess-23-4077-2019].

The same two species of shrubs are studied, and the field observations analysed here come from the same 2014-2015 data collection as were analysed by Yuan et al. (2019). The field data collection appears to have been one and the same for both papers.

Reply: *We acknowledge that this work and Yuan et al. (2019) did analyze the same two species of shrubs in the same area during 2014-2015, but our dataset was more complete, including data collection on throughfall and stemflow, as well as interception loss based on water balance. Yuan et al. (2019) only studied the branch-scale stemflow (only one of the rainfall partitioning processes), which can be seen from the title of Yuan et al. (2019). The research object of this work was individual shrub rather than branches, and the aim was to investigate in-deep the holistic processes of rainfall partitioning (i.e., throughfall, stemflow, and interception loss) at intra-event scales, for two typical xerophytic shrubs and to add some new insights into rainfall partitioning. To our best knowledge, this study is the first time to investigate the intra-event variations of the whole rainfall partitioning components for shrubs, on a single plant scale. This is the major novelty and advancement of this study. We have obtained the quantitative relationship between the rainfall partitioning variables and rainfall characteristics, and further elaborated the influences of vegetation structure characteristics (leaf, canopy structure, and biomass, etc..) on rainfall partitioning in the Discussion section.*

2. Comment:

Both Yuan et al. (2019) and the present ms. (An et al.) seek to explore the role of rainfall variability and plant architecture on stemflow, throughfall, and interception, paying attention to how these work at intra-event timescales. Their ability to do this is however hampered by their reliance on rainfall observations that were aggregated and logged only every 10 minutes. This is hardly sufficient temporal resolution to permit analysis of time lags before the commencement of stemflow, and various other analyses that the authors seek to make.

Reply: *Yuan et al. (2019) only explored the branch stemflow variations at intra-event timescale, and this study investigate the intra-event variations of the whole rainfall partitioning components on the plant scale (including throughfall, stemflow, and interception loss).*

*In terms of temporal resolution, the rainfall partitioning data at intra-event scale was recorded every 0.2 mm using tipping bucket rain gauge (TBRG). The data of stemflow and throughfall volume and timing were automatically recorded at dynamic intervals between neighboring tips. The observations data were aggregated and recorded every 10 minutes to better reflect fluctuations in rainfall partitioning components. A series of indices reflecting the dynamic process of rainfall partitioning were obtained.*

**3. Comment:**

Given that the two papers explore the same shrub taxa in the same field area during the same two years (2014-2015), and that both explore the intra-event workings of stemflow and throughfall, I think that a key requirement is for the Introduction to make it clear and explicit how the present paper differs in scope and results from the earlier paper of Yuan et al. (2019). The earlier paper appears to have focussed more strongly on branch-scale mechanisms, but a clear distinction requires very careful reading and differences in data processing make it very difficult indeed to see what is new in the current ms.

Reply: *Yuan et al. (2019) only explored the branch stemflow variations at intra-event timescale, and this study investigate the intra-event variations of the whole rainfall partitioning components on the plant scale (including throughfall, stemflow, and interception loss). In the Introduction section, we have made clear the differences between the present paper and the earlier paper by Yuan et al. (2019). The innovation of this study was to investigate in-deep the holistic processes of whole rainfall partitioning components (throughfall, stemflow, and interception loss) of shrub plant at inter- and intra-event scales.*

**4. Comment:**

In particular, in their Introduction (and perhaps also in a covering letter to accompany their submission) the authors should highlight what can be learned about stemflow and throughfall in the two shrub taxa that was not already demonstrated by Yuan et al. 2019). I think that it would be helpful for the authors to compare and contrast what was learned by Yuan et al. (2019) and what similarities or differences emerge in the present study (An et al.).

Reply: *This is a good comment. Yuan et al. (2019) described the stemflow variability at the branch scale and quantified its relationship with rainfall characteristics on inter- and intra-event. However, they did not study the variability of throughfall (the largest component of rainfall partitioning) and ignored interception loss. This work further provided the dynamics of the whole rainfall partitioning including stemflow, throughfall, and interception loss at the individual plant scale. In the Introduction section, we highlight the differences between this study and Yuan et al. (2019), and discuss in detail the innovative and important implications of this study in the Discussion section. For the results of stemflow, the two papers were largely similar, both in describing stemflow processes and in quantifying their relationship to the rainfall characteristics within event.*

**5. Comment:**

In other respects the paper seems entirely routine, containing nothing new in method, theory, or argument. I do think that the authors should evaluate the adequacy f their data and field sampling, however. Do the 38 rainfall events in two years form a sufficiently large sample of events? Does sampling of just four branches (line 202) represent a sufficiently large sample? What is the evidence for this being the case? And only a single individual plant of each species was used to measure throughfall and stemflow dynamics (line 198). How was the single individual selected? Is a sample of one really sufficient to draw meaningful conclusions concerning the entire species, as the authors do? These matters and similar considerations should be discussed and critically evaluated. As the paper stands, the authors merely report that they studied only a single shrub of each species but provide no evaluation of whether this is a sufficient and representative sample. (In the same vein, the use of aggregated, 10-minute rainfall amounts warrants critical comment by the authors. The authors should also report fully and properly the characteristics of the rainfall events that they monitored. To judge from the data in Yuan et al. (2019) these were mostly rather brief - from one to a few hours. But the present ms. (An et al.) does not even mention the event duration (nor, for instance, whether the rain was during daylight hours or at night - which is surely relevant to evaporative losses and hence to interception amounts). All of this must be corrected in a revision to the current ms. Some evidence of the nature of just four rainfall events can be found in Figure 5 but this is hardly sufficient.

Reply: *First, this is the first comprehensive study of intra-event dynamics of rainfall partitioning components (i.e., throughfall, stemflow and interception loss) in shrubs. This is the main novelty and advance of this study. We have obtained the quantitative relationship between rainfall partitioning variables and rainfall characteristics, and further elaborated the influence of vegetation structure characteristics (leaf, canopy structure, and biomass, etc..) on rainfall partitioning.*

*Second, in terms of the number of rainfall events and sampling methods, we refer to studies by Magliano et al., 2019b, Whitworth-Hulse et al., 2020a, and Yang et al., 2019, which studied 22-40 rainfall events. So that we think 38 rainfall events in two years constitute a relatively sufficiently large event sample. And we added the characteristics of the rainfall events in the paper. The start and end time and duration of rainfall were added in Fig. 5 (see Fig. 5). We also pointed out the need for long-term observational studies in the Discussion section.*

*Third, we measured branch diameter (BD, mm), length (BL, cm), and angle (BA, o) of all 143 and 218 branches of C. korshinskii and S. psammophila. And the BD categories are defined as 0-10, 10-15, 15-20, and >20mm to ensure an appropriate number of branches within the category. On this basis, for each species, we selected six representative branches to characterize intra-event stemflow dynamics through strict selection conditions, including those branches be distributed across the four BD categories and that there was no crossover between the experimental branch and adjacent branches, no inflection point from the tip of the branch to the base, and easy to measure. Due to missing or incomplete data, four branches were finally identified for each species, located in each of the four BD categories to measure stemflow. Branch information is supplemented in the article. The single individual shrub was selected from the three representative shrub plants with similar crown heights and crown areas in each shrub species. It was selected to measure the dynamics of intra-event rainfall partitioning, mainly because of the high cost of equipment, and the difficulty of placing a lot of TBRGs. We compared the mean depth of throughfall and stemflow measured by TBRGs with the mean measured manually and found no significant difference between the two methods. In the future experiment, we will add more branches and individual shrubs to expand the sample size.*

*Finally, we have explained that rainfall observations were summarized and recorded every 10 minutes to better reflect fluctuations in rainfall partitioning rather than the accuracy of our data recording. The accuracy of our data logging is recorded every 0.2 mm.*

Table 3. The generation thresholds and mean depth of rainfall partitioning by *C. korshinskii* and *S. psammophila* at inter- and intra-event scales. Values are mean ± SD.

| Species and rainfall partitioning | | The generation thresholds (mm) | | Average depth (mm) | |
|---|---|---|---|---|---|
| | | Inter-event | Intra-event | Inter-event | Intra-event |
| *C. korshinskii* | TF | 0.8 | 0.4 ± 0.2 | 8.1 ± 7.9 | 7.2 ± 5.4 |
| | SF | 1.7 | 1.0 ± 0.7 | 1.2 ± 1.3 | 1.3 ± 1.1 |
| *S. psammophila* | TF | 1.1 | 0.3 ± 0.1 | 8.3 ± 7.4 | 7.5 ± 5.7 |
| | SF | 2.2 | 0.7 ± 0.3 | 0.5 ± 0.5 | 0.9 ± 0.8 |

At inter-event scale, the thresholds for TF and SF generation are derived from the regression equation in Fig. 4. At intra-event scale the threshold values are measured using tipping bucket rain gauges.

[Figure]

**Figure 5.** Time series (10-min interval) of rainfall partitioning within four rainfall events for *C. korshinskii* (CK) and *S. psammophila* (SP). Events 1-4 occurred on August 3, September 17, September 28, and September 30 in 2015, respectively. The solid lines represent the rainfall, TF and SF intensity at 10-min interval. The dotted lines indicate the accumulated amount of RA, TF, SF, and IC.

Response to Anonymous Referee #2:

General comments:

Rainfall partitioning (interception loss, throughfall, and stemflow) is an old theme in the field of forest hydrology, and numerous studies have been done on quantified rainfall partitioning and their influencing factors at event scale. Nevertheless, the relevant studies are lacking for shrubs of drylands, and the intra-event dynamics have been less explored. An et al. characterized and quantified the rainfall partitioning of two xerophytic shrubs at both inter-event and intra-event scale in the Loess Plateau of China. What's really interesting to me is their concurrent finer investigation (10 min) on the intertwined rainfall partitioning processes. It seems that another paper by some of the same authors has been published in HESS (Yuan et al., 2019, 23(10): 4077-4095) in digging into the branch- scale dynamics of stemflow (only one of the rainfall partitioning processes). In my view, this study steps further and provides a full view of the reciprocal dynamics among interception loss, throughfall, and stemflow at the shrub-scale and thereby discussed the underlying mechanisms. In this sense, this study adds some new insights into rainfall partitioning and has the potential for a better understanding of the shrub-dominated eco- hydrological processes in drylands. Of course, the authors should explicitly explain the difference between two papers. Moreover, this paper is in general well-written; the experimental design and data analysis are normal and acceptable; results and discussion are informative. I have some moderate/minor comments that are required before considering the manuscript for publication.

Reply: *Thank you for nice comments and insightful suggestions. Yes, Yuan et al. (2019) described the branch-scale stemflow (only one of the rainfall partitioning processes), they did not examine throughfall (the largest component of rainfall), and ignored interception loss. This study combined the inter-event and intra-event variabilities of stemflow, throughfall, and interception loss at the individual plant scale to provide a further integrated analysis of overall rainfall partitioning dynamics. To our best knowledge, this study is the first time to investigate the intra-event variations of the whole rainfall partitioning components for shrubs. This is the major novelty and advancement of this study. In the Introduction and Discussion sections, we have clearly pointed out the differences and innovations of this study with respect to previous studies.*

Specific comments:

1. Comment:

L49: Water loss due to interception evaporates but not transpires back to atmosphere. "transpiration" here is NOT a correct term for interception loss.

Reply: *We have replaced "transpiration" with "evaporation" (see P.3, Lines 50).*

2. Comment:
L133: A citation is missed for Flora of China.

Reply: *We have added citation for Flora of China.*

3. Comment:

L271: What does values such as "11.1 ± 8 mm"? Mean ± SD or Mean ± SE? Better explain for the first time as they appear.

Reply: *It means Mean ± SD.*

4. Comment:

L305-311: Are the thresholds for the generation of TF and SF derived from regression equations comparable to or in the range of that measured using tipping bucket rain gage? A brief discussion somewhere in the Discussion section is desirable.

Reply: *We have compared the thresholds for the generation of TF and SF, derived from regression equations and measured using tipping bucket rain gauges, respectively (see Table 3) and discussed them. We found that TF and SF thresholds measured using the TBRGs were both smaller than the thresholds derived from the regression equation. This is a closer indication of the importance of high-resolution intra-event data in rainfall partitioning studies.*

Table 3. The generation thresholds and mean depth of rainfall partitioning by *C. korshinskii* and *S. psammophila* at inter- and intra-event scales. Values are mean ± SD.

| Species and rainfall partitioning | | The generation thresholds (mm) | | Average depth (mm) | |
|---|---|---|---|---|---|
| | | Inter-event | Intra-event | Inter-event | Intra-event |
| *C. korshinskii* | TF | 0.8 | 0.4 ± 0.2 | 8.1 ± 7.9 | 7.2 ± 5.4 |
| | SF | 1.7 | 1.0 ± 0.7 | 1.2 ± 1.3 | 1.3 ± 1.1 |
| *S. psammophila* | TF | 1.1 | 0.3 ± 0.1 | 8.3 ± 7.4 | 7.5 ± 5.7 |
| | SF | 2.2 | 0.7 ± 0.3 | 0.5 ± 0.5 | 0.9 ± 0.8 |

At inter-event scale, the thresholds for TF and SF generation are derived from the regression equation in Fig4. At intra-event scale the threshold values are measured using tipping bucket rain gauges.

5. Comment:

L370: significantly → significant?

Reply: *we have replaced "significantly" with "significant".*

6. Comment:

L404-414: The authors contributed the significant higher IC % in *C. korshinskii* than in *S. psammophila* to the higher water storage of *S. psammophila*. However, they used the absolute values (4.9 L versus 6.0 L) but not the normalized ones. Surely, a large canopy tends to absorb more rainwater than a small canopy, but that does necessarily mean that the large canopy has a higher capacity in retaining rainwater. Actually, the intercepts (0.92 for *C. korshinskii* and 1.15 for *S. psammophila*) in the fitted formulas between interception loss (mm) and rainfall amount (mm) in Figure 4e are indicative that *C. korshinskii* has a lower canopy water storage, hence a potential lower interception loss.

Reply: *The IC% of S. psammophila is significantly higher than that of C. korshinskii. This study was done at the shrub-scale, so we compared the total canopy water capacity of individual plant (Cm). While comparing Cm of S. psammophila (6.0 L) and C. korshinskii (3.9 L), we also discussed the intercepts (0.92 mm for C. korshinskii and 1.15 mm for S. psammophila) in the fitted formulas between*

*interception loss (mm) and rainfall amount (mm) in Figure 4e to further confirm the lower canopy water storage of C. korshinskii.*

7. Comment:

L443-447: I would like to argue with the authors that the dynamic characteristics for TF such as TFI, $TFI_{10}$, $LE_{TF}$, TFD, $LG_{TF}$ and $LM_{TF}$ are just different variables indicating the behaviors of TF but not the reasons for the generation of TF. Those variables are results but not the reasons. That means, according to those variables, it is reasonable to say which variables are indicative of an earlier or later generation of TF, but they are not the reasons that a shrub species is beneficial to the generation of TF. This is also the case for SF.

Reply: *The dynamic characteristics for TF such as TFI, $TFI_{10}$, $LE_{TF}$, TFD, $LG_{TF}$ and $LM_{TF}$ are variables that are indeed the result of TF production rather than the cause of TF production. But from these results we can also judge that those TF variables of S. psammophila can produce more throughfall depth (TFI\*TFD=TFd) than C. korshinskii. We therefore conclude that S. psammophila is more conducive to the generation of TF than C. korshinskii. The same is true for SF.*

8. Comment:

The authors made a detailed description of intra-event rainfall partitioning dynamics. I suggest that they elaborate more on the potential ecological implications.

Reply: *We have elaborated more on the potential ecological implications of intra-event rainfall partitioning dynamics in the Discussion section.*

**Response to Anonymous Referee #3:**

General comments:

The present work collected very detailed data to conduct a concurrent and in-depth investigation of throughfall (TF), stemflow (SF), and interception loss (IC) at both inter- and intra-event scales for two typical xerophytic shrubs on the dry region in the Chinese Loess Plateau, and the effects of bio-/abiotic factors were investigated. Previous publications from some of the same authors (Yuan et al., 2019, HESS) and the other researchers (Zhang et al., 2018, Science of The Total Environment; Yang et al., 2019, Journal of Hydrology etc.) only focused on TF or SF in shrubs, and the most of rainfall partitioning investigations are limited at inter-event scales. The intra-event rainfall partitioning dynamics which could help have a better understanding of soil water replenishment and its distribution in soil and the key ecohydrological cycle in arid regions have been rarely explored. As far as I know, this study is the first time to investigate the intra-event variations of all the rainfall partitioning components (i.e., TF, SF and IC) for shrubs. This is the main novelty and a step forward compared with the previous related studies (Yuan et al., 2019, HESS; Yuan et al., 2017, HESS; Yang et al., 2019, JH). This study obtained new insights to understand the fine characterization of shrub-dominated eco- hydrological processes, and improve the accuracy of water balance estimation in dryland ecosystem. The paper is well written

and interesting to the general readers of HESS, and I think it can be published in HESS. I have the following comments to further improve it.

Reply: *Thank you for the nice comments on the novelty of this study. This study combined the inter-event and intra-event variabilities of stemflow, throughfall, and interception loss to provide a full view of rainfall partitioning dynamics at the shrub-scale, and discussed the effects of rainfall and vegetation characteristics on rainfall partitioning dynamics.*

Specific comments:

1. Comment:

The authors should explain explicitly the novelty of this study, especially how it advances from Yuan et al. (2019) and Zhang et al. (2018).

Reply: *This study combined the inter-event and intra-event variabilities of stemflow, throughfall, and interception loss to provide a full view of rainfall partitioning dynamics at the shrub-scale. Most of previous studies on rainfall partitioning investigations of shrubs were limited at inter-event scale, or only focused on the single process (TF or SF) at the intra-event scale. In the Introduction section, we have clarified the differences between this paper and earlier papers by Yuan et al. (2019) and Zhang et al. (2018), and explicitly explained the progress made in this study in the discussion section.*

2. Comment:

Compare your stemflow data with that reported by Yang et al. (2019) in Journal of Hydrology for the same shrub species.

Reply: *We have compared our stemflow data with those for the same shrub species reported by Yang et al. (2019).*

3. Comment:

The authors selected three representative shrub plants to investigate inter-event rainfall partitioning. Eight TF manual gauges were placed under each *korshinskii* plant, and for *S. psammophila*, twenty TF gauges were placed under each plant. For SF yield, a total of 53 branches of *C. korshinskii* and 98 branches of *S. psammophila* were used. Compared to the thorough measurements at inter-event scale, the measurements at intra-event scale were somewhat limited (four TBRGs (tipping bucket rain gauges) for intra-event TF, and four TBRGs for intra-event SF). I know it is mainly due to high cost of equipment, and it is difficult to place a lot of TBRGs to measure intra-event rainfall partitioning. The authors should discuss this issue.

Reply: *We measured all branches of C. korshinskii (143) and S. psammophila (218), including branch diameter (BD, mm), branch length (BL, cm) and branch angle (BA, º). Based on this, branches were divided into four categories and four representative branches were finally identified, with one branch per category selected to measure intra-event stemflow using TBRGs (6.7, 13.5, 18.6, and 22.1 mm for C. korshinskii and 7.2, 14.4, 18.2, and 31.3 mm for S. psammophila). Four TBRGs were placed in four directions to measure throughfall. Due to high cost of equipment, measurements at the intra-event scale did have limitations compared to the thorough measurements at the inter-event scale. In the*

*Discussion section, We also point out that more plants and branches need to be selected for long-term observation studies in the future.*

4. Comment:

Some newest references are lost from this paper, such as "Yue et al., 2021, Global patterns and drivers of rainfall partitioning by trees and shrubs, Global Change Biology". The authors should check it.

Reply: *We have added the newest and relevant references.*

5. Comment:

Whether the rain ended at daylight hours or at night? How long after the end of rainfall to collect throughfall in TF manual gauges? The effects of relevant evaporative losses in TF manual gauges should be discussed, as they are open to atmosphere as shown in Fig. 1b.

Reply: *If the rainfall ended during the day, we completed the data collection of throughfall (TF manual rain gauge) within two hours after the termination of rainfall. And if the rainfall ended at night, we completed the data collection as early as possible the following day to reduce the effects of evaporative losses.*

6. Comment:

Line 49, transpiration should be evaporation.

Reply: *Transpiration has been changed to evaporation.*

7. Comment

While describing the intra-event rainfall partitioning dynamics, the authors should elaborate more on its potential ecological significance.

Reply: *The potential ecological significance of the intra-event rainfall partitioning dynamics has been described in the Discussion section.*

8. Comment

The authors did not express clearly whether the 38 rainfall events were all rainfall events in 2014-2015 rainy seasons or those producing throughfall and stemflow.

Reply: *We have clarified that these 38 rainfall events are the effective rainfall events that produced throughfall and stemflow during the 2014-2015 rainy season, and not all rainfall events.*

9. Comment

The authors should describe the relationships between intra-event rainfall partitioning variables and meteorological factors such as wind speed and wind direction, even there were no significant relevance.

Reply: *The intra-event rainfall partitioning variables were not significantly correlated with meteorological factors such as wind speed and direction, which has been described.*

**10. Comment**

The possible limitations of your study and the future research focus are suggested to be included in the final section the Discussion part.

Reply: *In the Discussion section, we have discussed the limitations of the experimental design and further scopes of this study. An important research topic is to combine the dynamic process of rainfall partitioning with the soil moisture and evapotranspiration responses to systematically explain the complete eco-hydrological process and to portray the dynamics of this process over time.*

**11. Comment**

In some references, the authors' first and last name is incomplete. Please revise.

Reply: *Incomplete authors' first and last name have been revised in the references.*

---

## Author Response (AR1)

May 1, 2022

Memorandum

To:        Pro. Lixin Wang, Editor of *Hydrology and Earth System Sciences*

Subject:    **Revision of hess-2022-4**

Dear Pro. Wang:

Upon your request, we have carefully addressed all the comments made by David Dunkerley and the two anonymous reviewers on our manuscript (hess-2022-4) entitled "Inter- and intra-event rainfall partitioning dynamics of two typical xerophytic shrubs in the Loess Plateau of China" and revised the manuscript accordingly. The comments have helped us greatly improve the overall quality of the manuscript. The page and line numbers in the following response refer to the revised manuscript with changes marked.

Response to Referee #1 (Pro. David Dunkerley):

1.   Comment:

This paper explores a little further data from the same field observations as were discussed previously by some of the same authors in their 2019 HESS paper *Temporally dependent effects of rainfall characteristics on inter- and intra-event branch-scale stemflow variability in two xerophytic shrubs* (Yuan et al. 2019).

[Yuan, C., Gao, G., Fu, B., He, D., Duan, X., & Wei, X. (2019). *Temporally dependent effects of rainfall characteristics on inter- and intra-event branch-scale stemflow variability in two xerophytic shrubs. Hydrol. Earth Syst. Sci.*, 23(10), 4077-4095. doi:10.5194/hess-23-4077-2019].

The same two species of shrubs are studied, and the field observations analysed here come from the same 2014-2015 data collection as were analysed by Yuan et al. (2019). The field data collection appears to have been one and the same for both papers.

Reply: *We acknowledge that this work and Yuan et al. (2019) did analyze the same two species of shrubs in the same area during 2014-2015, but our dataset was more complete, including data collection on throughfall and stemflow, as well as interception loss based on water balance. Yuan et al. (2019) only studied the branch-scale stemflow (only one of the rainfall partitioning processes), which can be seen from the title of Yuan et al. (2019) (see P.5, Lines 109-116). The research object of this work was individual shrub rather than branches, and the aim was to investigate in-deep the holistic processes of rainfall partitioning (i.e., throughfall, stemflow, and interception loss) at both inter-event and intra-event scales, and to add some new insights into rainfall partitioning (see P.6, Lines 130-131). To our best knowledge, this study is the first time to investigate the intra-event variations of the whole rainfall partitioning components for shrubs. This is the major novelty and advancement of this study. We have also obtained the quantitative relationships between the rainfall partitioning variables and rainfall characteristics, and further elaborated the influences of vegetation structure characteristics (leaf,*

*canopy structure, and biomass, etc..) on rainfall partitioning. This study obtained new insights to understand the fine characterization of shrub-dominated eco- hydrological processes, and improve the accuracy of water balance estimation in dryland ecosystem (see P.24, Line 556 to P.25, Line 578).*

2. Comment:

Both Yuan et al. (2019) and the present ms. (An et al.) seek to explore the role of rainfall variability and plant architecture on stemflow, throughfall, and interception, paying attention to how these work at intra-event timescales. Their ability to do this is however hampered by their reliance on rainfall observations that were aggregated and logged only every 10 minutes. This is hardly sufficient temporal resolution to permit analysis of time lags before the commencement of stemflow, and various other analyses that the authors seek to make.

Reply: *Yuan et al. (2019) only explored the variations of branch stemflow, and this study investigate the inter- and intra-event variations of the whole rainfall partitioning components on the shrub scale (including throughfall, stemflow, and interception loss). (see P.5, Lines 109-116; P.6, Lines 130-131)*

*In terms of temporal resolution, the rainfall partitioning data at intra-event scale was recorded every 0.2 mm using tipping bucket rain gauge (TBRG). The data of stemflow and throughfall volume and timing were automatically recorded at dynamic intervals between neighboring TBRG tips. The observations data were aggregated every 10 minutes to better reflect fluctuations in rainfall partitioning components (see P.12, Lines 275-278). A series of indices reflecting the dynamic processes of rainfall partitioning were obtained (see P.13, Lines 295-304).*

3. Comment:

Given that the two papers explore the same shrub taxa in the same field area during the same two years (2014-2015), and that both explore the intra-event workings of stemflow and throughfall, I think that a key requirement is for the Introduction to make it clear and explicit how the present paper differs in scope and results from the earlier paper of Yuan et al. (2019). The earlier paper appears to have focussed more strongly on branch-scale mechanisms, but a clear distinction requires very careful reading and differences in data processing make it very difficult indeed to see what is new in the current ms.

Reply: *Yuan et al. (2019) only explored the branch stemflow variations at intra-event timescale, and this study investigate the intra-event variations of the whole rainfall partitioning components on the shrub scale (including throughfall, stemflow, and interception loss). In the Introduction section, we have made clear the differences between the present paper and the earlier paper by Yuan et al. (2019). (see P.5, Lines 109-116; P.6, Lines 130-131)*

*The innovation of this study was to investigate in-deep the holistic processes of whole rainfall partitioning components (throughfall, stemflow, and interception loss) of shrub plant at inter- and intra-event scales. We have also obtained the quantitative relationships between the rainfall partitioning variables and rainfall characteristics, and further elaborated the influences of vegetation structure characteristics (leaf, canopy structure, and biomass, etc..) on rainfall partitioning. (see P.24, Line 556 to P.25, Line 578).*

4. Comment:

In particular, in their Introduction (and perhaps also in a covering letter to accompany their submission) the authors should highlight what can be learned about stemflow and throughfall in the two shrub taxa that was not already demonstrated by Yuan et al. 2019). I think that it would be helpful for the authors to compare and contrast what was learned by Yuan et al. (2019) and what similarities or differences emerge in the present study (An et al.).

Reply: *Yuan et al. (2019) described the stemflow variability at the branch scale, and quantified its relationship with rainfall characteristics on inter- and intra-event scales. However, they did not study the variability of throughfall (the largest component of rainfall partitioning) and ignored interception loss. This work further provided the dynamics of the whole rainfall partitioning including stemflow, throughfall, and interception loss at the individual plant scale. In the Introduction section, we have highlighted the differences between this study and Yuan et al. (2019). (see P.5, Lines 109-116; P.6, Lines 130-131) The novelty and implications of this study have been discussed in detail in the Discussion section (see P.24, Line 556 to P.25, Line 578). For the results of stemflow, the two papers were largely similar, both in describing stemflow processes and in quantifying the relationships with the rainfall characteristics (see P.22, Lines 508-510; P.23, Lines 522-531; P.24, Lines 543-545).*

5. Comment:

In other respects the paper seems entirely routine, containing nothing new in method, theory, or argument. I do think that the authors should evaluate the adequacy f their data and field sampling, however. Do the 38 rainfall events in two years form a sufficiently large sample of events? Does sampling of just four branches (line 202) represent a sufficiently large sample? What is the evidence for this being the case? And only a single individual plant of each species was used to measure throughfall and stemflow dynamics (line 198). How was the single individual selected? Is a sample of one really sufficient to draw meaningful conclusions concerning the entire species, as the authors do? These matters and similar considerations should be discussed and critically evaluated. As the paper stands, the authors merely report that they studied only a single shrub of each species but provide no evaluation of whether this is a sufficient and representative sample. (In the same vein, the use of aggregated, 10-minute rainfall amounts warrants critical comment by the authors. The authors should also report fully and properly the characteristics of the rainfall events that they monitored. To judge from the data in Yuan et al. (2019) these were mostly rather brief - from one to a few hours. But the present ms. (An et al.) does not even mention the event duration (nor, for instance, whether the rain was during daylight hours or at night - which is surely relevant to evaporative losses and hence to interception amounts). All of this must be corrected in a revision to the current ms. Some evidence of the nature of just four rainfall events can be found in Figure 5 but this is hardly sufficient.

Reply: *First, this is the first comprehensive study to investigate the intra-event dynamics of all rainfall partitioning components (i.e., throughfall, stemflow and interception loss) in shrubs. This is the main novelty and advance of this study. We have also obtained the quantitative relationship between rainfall partitioning variables and rainfall characteristics, and further elaborated the influence of vegetation structure characteristics (leaf, canopy structure, and biomass, etc..) on rainfall partitioning (see P.24, Line 556 to P.25, Line 578).*

*Second, in terms of the number of rainfall events, most of previous studies such as Magliano et al., 2019b, Whitworth-Hulse et al., 2020a, and Yang et al., 2019, only studied 22-40 rainfall events. So that we think 38 rainfall events in two years constitute a relatively sufficiently large event sample. We have*

*added the characteristics of the rainfall events in the revision version (see P.14, Line 321 to P.15, Line 337). The start and end time and duration of rainfall events were added in Fig. 5. We have also pointed out the need for long-term observational studies in the Discussion section (see P.25, Lines 581-583).*

*Third, we measured branch diameter (BD, mm), length (BL, cm), and angle (BA, º) of all 143 and 218 branches of four C. korshinskii and S. psammophila shrubs. Four BD categories were defined as 0-10, 10-15, 15-20, and >20 mm to ensure an appropriate number of branches within the category. For each species, we selected six representative branches to characterize intra-event stemflow dynamics through strict selection conditions, including those branches be distributed across the four BD categories and that there was no crossover between the experimental branch and adjacent branches, no inflection point from the tip of the branch to the base. Due to missing or incomplete data, four branches were finally identified for each species, located in each of the four BD categories to measure stemflow (see P.10, Line 235 to P.11, Line 246). The single individual shrub was selected from the three representative shrub plants in each shrub species to measure the dynamics of intra-event rainfall partitioning. Due to the high cost of equipment, it is difficulty to place a lot of TBRGs. In the future experiment, we will add more branches and individual shrubs to expand the sample size (see P.25, Lines 581-583).*

*Finally, we have explained that the rainfall partitioning data at intra-event scale was recorded every 0.2 mm using tipping bucket rain gauge (TBRG), and they were aggregated every 10 minutes to better reflect fluctuations in rainfall partitioning (see P.12, Lines 275-278).*

**Response to Anonymous Referee #2:**

**General comments:**

Rainfall partitioning (interception loss, throughfall, and stemflow) is an old theme in the field of forest hydrology, and numerous studies have been done on quantified rainfall partitioning and their influencing factors at event scale. Nevertheless, the relevant studies are lacking for shrubs of drylands, and the intra-event dynamics have been less explored. An et al. characterized and quantified the rainfall partitioning of two xerophytic shrubs at both inter-event and intra-event scale in the Loess Plateau of China. What's really interesting to me is their concurrent finer investigation (10 min) on the intertwined rainfall partitioning processes. It seems that another paper by some of the same authors has been published in HESS (Yuan et al., 2019, 23(10): 4077-4095) in digging into the branch- scale dynamics of stemflow (only one of the rainfall partitioning processes). In my view, this study steps further and provides a full view of the reciprocal dynamics among interception loss, throughfall, and stemflow at the shrub-scale and thereby discussed the underlying mechanisms. In this sense, this study adds some new insights into rainfall partitioning and has the potential for a better understanding of the shrub-dominated eco- hydrological processes in drylands. Of course, the authors should explicitly explain the difference between two papers. Moreover, this paper is in general well-written; the experimental design and data analysis are normal and acceptable; results and discussion are informative. I have some moderate/minor comments that are required before considering the manuscript for publication.

Reply: *Thank you for nice comments and insightful suggestions. Yes, Yuan et al. (2019) described the branch-scale stemflow (only one of the rainfall partitioning processes), and they did not examine*

*throughfall (the largest component of rainfall), and ignored interception loss. This study combined the inter-event and intra-event variabilities of stemflow, throughfall, and interception loss at the individual plant scale to provide a further integrated analysis of overall rainfall partitioning dynamics. To our best knowledge, this study is the first time to investigate the intra-event variations of the whole rainfall partitioning components for shrubs. This is the major novelty and advancement of this study. In the Introduction and Discussion sections, we have clearly pointed out the differences and innovations of this study with respect to previous studies (see P.5, Lines 109-116; P.6, Lines 130-131; P.24, Line 556 to P.25, Line 578).*

Specific comments:

1. Comment:

L49: Water loss due to interception evaporates but not transpires back to atmosphere. "transpiration" here is NOT a correct term for interception loss.

Reply: *We have replaced "transpiration" with "evaporation" (see P.3, Line 49).*

2. Comment:

L133: A citation is missed for Flora of China.

Reply: *We have added citation for Flora of China (see P.7, Lines 160-161).*

3. Comment:

L271: What does values such as "11.1 ± 8 mm"? Mean ± SD or Mean ± SE? Better explain for the first time as they appear.

Reply: *It means Mean ± SD (see P.14, Line 325).*

4. Comment:

L305-311: Are the thresholds for the generation of TF and SF derived from regression equations comparable to or in the range of that measured using tipping bucket rain gage? A brief discussion somewhere in the Discussion section is desirable.

Reply: *We have compared the thresholds for the generation of TF and SF, derived from regression equations and measured using tipping bucket rain gauges, and discussed the reason. We found that TF and SF thresholds measured using the TBRGs were both smaller than the thresholds derived from the regression equation. This is a closer indication of the importance of high-resolution intra-event data in rainfall partitioning studies (see P.17, Line 392 to P.18, Line 401).*

5. Comment:

L370: significantly → significant?

Reply: *we have replaced "significantly" with "significant" (see P.19, Line 441).*

6. Comment:

L404-414: The authors contributed the significant higher IC % in *C. korshinskii* than in *S. psammophila* to the higher water storage of *S. psammophila*. However, they used the absolute values (4.9 L versus 6.0 L) but not the normalized ones. Surely, a large canopy tends to absorb more rainwater than a small canopy, but that does necessarily mean that the large canopy has a higher capacity in retaining rainwater. Actually, the intercepts (0.92 for *C. korshinskii* and 1.15 for *S. psammophila*) in the fitted formulas between interception loss (mm) and rainfall amount (mm) in Figure 4e are indicative that *C. korshinskii* has a lower canopy water storage, hence a potential lower interception loss.

Reply: *The IC% of S. psammophila is significantly higher than that of C. korshinskii. This study was done at the shrub-scale, so we compared the total canopy water capacity of individual plant (Cm). While comparing Cm of C. korshinskii (3.9 L) and S. psammophila (6.0 L), we also discussed the intercepts (0.92 mm for C. korshinskii and 1.15 mm for S. psammophila) in the fitted formulas between interception loss (mm) and rainfall amount (mm) in Figure 4e to further confirm the lower canopy water storage of C. korshinskii (see P.21, Line 482 to P.22, Line 495).*

7. Comment:

L443-447: I would like to argue with the authors that the dynamic characteristics for TF such as TFI, $TFI_{10}$, $LE_{TF}$, TFD, $LG_{TF}$ and $LM_{TF}$ are just different variables indicating the behaviors of TF but not the reasons for the generation of TF. Those variables are results but not the reasons. That means, according to those variables, it is reasonable to say which variables are indicative of an earlier or later generation of TF, but they are not the reasons that a shrub species is beneficial to the generation of TF. This is also the case for SF.

Reply: *Yes, this is a good comment. The dynamic characteristics for TF such as TFI, $TFI_{10}$, $LE_{TF}$, TFD, $LG_{TF}$ and $LM_{TF}$ indicated the behaviors of TF production rather than the reasons for the generation of TF production. This is also the case for SF. We have deleted the confusing statement ("beneficial to the generation of TF and SF") in this sentence. (see P.24, Lines 537-541)*

8. Comment:

The authors made a detailed description of intra-event rainfall partitioning dynamics. I suggest that they elaborate more on the potential ecological implications.

Reply: *We have elaborated more on the potential ecological implications of intra-event rainfall partitioning dynamics in the Discussion section (see P.22, Line 510 to P.23, Line 518; P.25, Lines 576-578).*

Response to Anonymous Referee #3:

General comments:

The present work collected very detailed data to conduct a concurrent and in-depth investigation of throughfall (TF), stemflow (SF), and interception loss (IC) at both inter- and intra-event scales for two typical xerophytic shrubs on the dry region in the Chinese Loess Plateau, and the effects of bio-/abiotic factors were investigated. Previous publications from some of the same authors (Yuan et al., 2019, HESS) and the other researchers (Zhang et al., 2018, Science of The Total Environment; Yang et al., 2019, Journal of Hydrology etc.) only focused on TF or SF in shrubs, and the most of rainfall partitioning investigations are limited at inter-event scales. The intra-event rainfall partitioning dynamics which could help have a better understanding of soil water replenishment and its distribution in soil and the key ecohydrological cycle in arid regions have been rarely explored. As far as I know, this study is the first time to investigate the intra-event variations of all the rainfall partitioning components (i.e., TF, SF and IC) for shrubs. This is the main novelty and a step forward compared with the previous related studies (Yuan et al., 2019, HESS; Yuan et al., 2017, HESS; Yang et al., 2019, JH). This study obtained new insights to understand the fine characterization of shrub-dominated eco- hydrological processes, and improve the accuracy of water balance estimation in dryland ecosystem. The paper is well written and interesting to the general readers of HESS, and I think it can be published in HESS. I have the following comments to further improve it.

Reply: *Thank you for the nice comments on the novelty of this study. This study combined the inter-event and intra-event variabilities of stemflow, throughfall, and interception loss to provide a full view of rainfall partitioning dynamics at the shrub-scale, and discussed the effects of rainfall and vegetation characteristics on rainfall partitioning dynamics.*

Specific comments:

1. Comment:

The authors should explain explicitly the novelty of this study, especially how it advances from Yuan et al. (2019) and Zhang et al. (2018).

Reply: *This study combined the inter-event and intra-event variabilities of stemflow, throughfall, and interception loss to provide a full view of rainfall partitioning dynamics at the shrub-scale. Most of previous studies on rainfall partitioning investigations of shrubs were limited at inter-event scale, or only focused on the single process (TF or SF) at the intra-event scale. In the Introduction section, we have clarified the differences between this paper and earlier papers by Yuan et al. (2019) and Zhang et al. (2018) (see P.5, Lines 109-116; P.6, Lines 130-131). The novelty and implications of this study have been discussed in detail in the Discussion section (see P.24, Line 556 to P.25, Line 578).*

2. Comment:

Compare your stemflow data with that reported by Yang et al. (2019) in Journal of Hydrology for the same shrub species.

Reply: *We have compared our stemflow data with those for the same shrub species reported by Yang et al. (2019). (see P.20, Lines 448-450)*

3. Comment:

The authors selected three representative shrub plants to investigate inter-event rainfall partitioning. Eight TF manual gauges were placed under each *korshinskii* plant, and for *S. psammophila*, twenty TF gauges were placed under each plant. For SF yield, a total of 53 branches of *C. korshinskii* and 98 branches of *S. psammophila* were used. Compared to the thorough measurements at inter-event scale, the measurements at intra-event scale were somewhat limited (four TBRGs (tipping bucket rain gauges) for intra-event TF, and four TBRGs for intra-event SF). I know it is mainly due to high cost of equipment, and it is difficult to place a lot of TBRGs to measure intra-event rainfall partitioning. The authors should discuss this issue.

Reply: *We measured characteristics of all branches of the selected C. korshinskii and S. psammophila shrubs, including branch diameter (BD, mm), branch length (BL, cm) and branch angle (BA, º). Based on the measured BD, branches were divided into four categories (BD: 0-10, 10-15, 15-20, and >20 mm), and one branch per each category was selected to measure intra-event stemflow using TBRGs (BD: 6.7, 13.5, 18.6, and 22.1 mm for C. korshinskii and 7.2, 14.4, 18.2, and 31.3 mm for S. psammophila). Four TBRGs were placed in four directions to measure intra-event throughfall (see P.10, Line 235 to P.11, Line 246). Due to high cost of equipment, measurements at the intra-event scale did have limitations compared to the thorough measurements at the inter-event scale. In the Discussion section, we have also point out that more plants and branches need to be selected for long-term observation studies in the future (see P.25, Lines 581-583).*

**4. Comment:**

Some newest references are lost from this paper, such as "Yue et al., 2021, Global patterns and drivers of rainfall partitioning by trees and shrubs, Global Change Biology". The authors should check it.

Reply: *We have added some newest references (see P.3, Line 64).*

**5. Comment:**

Whether the rain ended at daylight hours or at night? How long after the end of rainfall to collect throughfall in TF manual gauges? The effects of relevant evaporative losses in TF manual gauges should be discussed, as they are open to atmosphere as shown in Fig. 1b.

Reply: *If the rainfall ended during the day, we completed the data collection of throughfall (TF manual rain gauge) within two hours after the termination of rainfall. And if the rainfall ended at night, we completed the data collection as early as possible in the following day to reduce the effects of evaporative losses (see P.10, Lines 220-223).*

**6. Comment:**

Line 49, transpiration should be evaporation.

Reply: *Transpiration has been changed to evaporation (see P.3, Line 49).*

**7. Comment**

While describing the intra-event rainfall partitioning dynamics, the authors should elaborate more on its potential ecological significance.

Reply: *The potential ecological significance of the intra-event rainfall partitioning dynamics has been described in the Discussion section (see P.22, Line 510 to P.23, Line 518; P.25, Lines 576-578).*

**8. Comment**

The authors did not express clearly whether the 38 rainfall events were all rainfall events in 2014-2015 rainy seasons or those producing throughfall and stemflow.

Reply: *We have clarified that these 38 rainfall events are the effective rainfall events that produced throughfall and stemflow during the 2014-2015 rainy season, and not all rainfall events (see P.14, Lines 321-324).*

**9. Comment**

The authors should describe the relationships between intra-event rainfall partitioning variables and meteorological factors such as wind speed and wind direction, even there were no significant relevance.

Reply: *The intra-event rainfall partitioning variables were not significantly correlated with meteorological factors such as wind speed and direction, which has been described in the revised version (see P.19, Lines 431-432).*

**10. Comment**

The possible limitations of your study and the future research focus are suggested to be included in the final section the Discussion part.

Reply: *This is a good suggestion. In the Discussion section, we have discussed the limitations of the experimental design and further scopes of this study (see P.25, Lines 579-588).*

**11. Comment**

In some references, the authors' first and last name is incomplete. Please revise.

Reply: *Incomplete authors' first and last name have been revised in the references (see "References", P.27, Line 622 to P.35, Line 789).*

---

## Author Response (AR2)

Memorandum

To:       Pro. Lixin Wang, Editor of *Hydrology and Earth System Sciences*

Subject:   **Revision of hess-2022-4**

**Response to Editor:**

Dear authors,

Thank you for the thorough revision of the manuscript. All three reviewers are generally satisfied with the revision and supported the publication of this manuscript. At the same time, one reviewer pointed out several important issues that require further justification or discussion. I look forward to reading a revised version of this manuscript with the reviewer's additional comments in mind. Thank you!

Yours sincerely,

Lixin

Reply: *Thank you for nice comments on our revision work. Pro. David Dunkerley provided some further comments, and the other two reviewers agreed to accept the paper. We have carefully addressed all the comments and revised the manuscript. The comments have helped us further improve the overall quality of the manuscript. The page and line numbers in the following response refer to the revised manuscript with changes marked.*

   *In addition, Pro. Juan Pinos helped us polish the manuscript, and we added him in the author list.*

**Response to Referee #1 (Pro. David Dunkerley):**

**General comments:**

This paper analyses some field data on interception, throughfall, and stemflow in two shrub taxa from the Chinese drylands. The data were collected in 2014-2015. The authors explore some of the behaviour of the interception parameters both among separate rainfalls and at the intra-event scale.

On the whole, the paper is very systematically set out, and generally clear. The work reflects a considerable and commendable effort in instrumenting dryland shrubs to record interception, stemflow, and throughfall. It is good to see such work carried out under natural rainfall, and not rainfall simulation, which generally fails to reproduce key characteristics of natural rainfall events, notably including their durations. Nevertheless, there are some more serious issues with the data and the interpretation of the results, which I explore further below.

Reply: *Thank you for nice comments on our revision work and providing further suggestions. We have addressed all the comments and incorporated them to revise the manuscript.*

1. Comment:

A fundamental issue that the authors do not dicuss is whether it is actually important to investigate water partitioning for individual shrubs (as they do in this paper) or whether area-wide water balance is more significant, given that shrub root systems can extend widely, well beyond the limits of the canopy. In other words, for a given shrub, is it not possible that an important component of the water accessed via the roots is actually from open-field rainfall, and not from stemflow or throughfall? A critical issue here is whether, and to what extent, shrubs can generate metabolically useful stemflow, say, focussed on the root system. Otherwise, shrub canopies do nothing but reduce the depth of rainfall arriving at the root system (owing to interception losses on the above-ground plant parts). This is why extensive root networks become so important. It is also generally considered that small showers of rain are not of great use to shrubs, many of which have deep root systems that access water accumulated in the deeper soil over months or years. Small rainfall events can nevertheless benefit microphytic plants and microbial communities.

The authors need not resolve the issue of individual shrub versus landscape-level analysis, but in their paper, they certainly need to discuss this, and show that they have considered what their results really tell us that is of ecological use, in helping to understand the true water balance of shrublands. For example, what is the actual ground cover fraction covered by shrub canopies, in their study area, and how much soil is exposed directly to rainfall with no interception?

Reply: *This is an insight comment. The study area is in the south fringe of Mu Us sandy land in North China, and the shrub is distributed sparsely with distinct interspaces. The actual ground cover fraction covered by shrub canopies was less than 20%, with the rest of soil being exposed directly to rainfall with no interception (see P.7, Lines 176-178). Therefore, this study focused on the rainfall partitioning (interception, throughfall, and stemflow) of individual shrub. We agree with the reviewer that the small rainfall events are very important for the xerophytic shrubs in drylands. We have measured the soil moisture, water isotopes and root systems. In our next research, we will analyze the soil moisture responses to rainfall partitioning and examine how the shrubs actually make use of small amounts of throughfall or stemflow (see P.26, Lines 615-619). It is also an important further scope to extend the work on individual shrub to the landscape-level analysis (see P.26, Lines 619-622).*

There are also technical issues that I think require further classification and justification.

2. Comment:

The authors derive an estimate of I10max. But as far as I can see, their rainfall data are locked to the time-step of the logger. To find short-term maximum intensities correctly, much higher temporal resolution of the rainfall data is needed. For instance, suppose the logger records the rainfall in the periods 10:00-10:10, 10:10-10:20, etc. Then, if the most intense 10 minutes actually occurred from 10:05-10:15, it will be missed by the logger data, and the maximum intensity recorded will only be about

half of the true value.

Reply: *In this study, to better reflect fluctuations in rainfall partitioning components at the intra-event scale, gross rainfall, TF and SF data measured by tipping-bucket rain gauge (with a 0.2 mm resolution) were all aggregated every 10 minutes. The $I_{10}$ means rainfall intensity at 10-min interval (i.e., 0-10 min, 10-20 min,....) since the start of rainfall. $I_{10\_max}$ means the maximum $I_{10}$ of one rainfall event, not maximum rainfall intensity in 10 minutes (see P.8, Lines 199-202).*

3. Comment:

The authors express canopy water holding capacity (C) by weight gain of branch specimens dipped in water. They then proceed to estimate the canopy storage capacity of entire shrubs by multiplying by the total biomass of the shrub, estimated from an allometric growth model. Why they did not actually weigh some entire shrubs is unclear. In any case, the problem with this procedure is that it is far more likely to be the surface area of the plant parts that governs C, and not weight. The authors need to discuss this, and to defend or justify the procedure that they adopted, based on mass not surface area.

Reply: *First, it is difficult to weigh the entire shrubs due to the high canopy volume. The average height and canopy area of S. psammophila is 3.5 m and 23.8 $m^2$, respectively, and the corresponding value of C. korshinskii is 2.3 m and 5.27 $m^2$ (see Table 1). The allometric growth model obtained in our previous work had very high accuracy with $R^2$ more than 0.92, which can be used to estimate the total biomass of the shrub (see P.9, Line 238 to P.10, Line 241).*
    *Second, the surface area of the plant parts is an important factor affecting canopy water holding capacity, and the mass of the plant also plays an important role. However, how to accurately obtain the surface area of the plant parts (including branch, stem and leaf) is a great challenge. In comparison, the weight of shrub can be easily and accurately obtained. Therefore, the water immersion method (i.e., based on mass) was widely used in previous studies to measure canopy water holding capacity (see P.9, Lines 231-236).*

4. Comment:

Likewise, the authors estimate total shrub stemflow by measurements on a few branches, then multiply the result by the total number of branches on the shrub. This again seems unlikely to be reliable, given that some branches form the outer perimeter of a shrub, and some are more sheltered, internal branches. There are also upper, more exposed branches, and lower, more sheltered branches within the canopy. The presumption that all of these generate the same stemflow flux warrants justification.

Reply: *Three representative shrub plants were selected in each shrub species to measure rainfall partitioning at inter-event scale. The total number of branches was 143 and 218 for selected C. korshinskii and S. psammophila plants, respectively. A total of 53 branches of C. korshinskii and 98 branches of S. psammophila were used to measure stemflow (37.06% and 44.95% of total number, not just a few branches), which covered different types of branches (see P.10, Lines 256-259). The SF volumes measured on the selected branches were averaged to obtain the average volume of SF on the branch scale. We multiplied the average branch SF with the number of branches to obtain the total SF volume from the plant, but did not make presumption that all of these generate the same stemflow flux.*

The maximum throughfall fractions reported by the authors - around 80% - seem very low in light of the depth of some of the rainfall events (more than 40 mm). I would expect that in such an event, the throughfall fraction would approach 100%.

Consider some rough calculations:

For both shrubs, the value of Cm is about 5 litre.

For C Korshinskii, the canopy area averages ~ 5 m2, so the depth of rain over the canopy required to fill Cm is about 1 mm.

For S psammophila, the canopy area averages ~ 24 m2, so the depth of rain over the canopy required to fill Cm is about 0.2 mm.

Both depths required to fill the canopy stores are negligible in relation to a rainfall of 40 mm, and should have resulted in an interception loss (even allowing for ongoing evaporation during rainfall) of at least 97%. Where did the rainwater go to yield throughfall fractions no larger than 80%? Here I begin to have serious doubts about the reliability of the field data. Perhaps the few throughfall gauges deployed underestimated that parameter? Perhaps intra-event evaporation is actually very significant for these shrubs? The authors need to comment on these possibilities and explain why the maximum throughfall fraction that they report is so low in events in which the loss to canopy storage should be no more than 1% or so of the incident rainfall.

Reply: *We think the reviewer ignores the stemflow, which is also an important component of rainfall partitioning. For the interception loss, besides the water storage capacity of the canopy, ongoing evaporation also accounts for a significant fraction.*

*In this study, we conducted solid field measurement work, as indicated by all the three reviewers. The maximum throughfall fractions of C. korshinskii and S. psammophila reached 79.2% and 80.0%, respectively. The maximum SF% approximately tended to be at 12.2% and 5.5% for C. korshinskii and S. psammophila, respectively. The maximum IC% approximately tended to be at 9.0% and 14.5% for C. korshinskii and S. psammophila, respectively (see P.17, Lines 410-421). The obtained results are comparable to those in previous studies (see P.20. Lines 487-489; P.22, Lines 517-522). It is a further scope to measure every component of shrub canopy water balance (see P.26, Lines 614-615).*

These queries lead me once again to recommend that the authors provide more detail on the rainfall events that they recorded. How were separate events delineated? How were intensity and duration related, for these events? Further, given the air temperature and humidity data that the authors had available, did they estimate ongoing wet canopy evaporation losses during rainfall, as a component of the shrub canopy water balance?

Reply: *We have provided the detailed characteristics of the rainfall events, including rainfall amount, intensity and duration in the results (see P.15, Lines 362-376). Although we have measured air temperature and humidity data using meteorological station, it is still difficult to estimate wet canopy evaporation loss which needs microclimate data. It is a further scope to measure every component of shrub canopy water balance (see P.26, Lines 614-615).*

**7. Comment:**

Finally, of course, some further data on the shrublands themselves would be of use. What was the areal cover provide by shrubs (as distinct from shrub interspaces)? What were the shrub root systems actually like, and can the shrubs actually make use of small amounts of throughfall or stemflow? Though arguing that these aspects of shrub hydrology are important for us to understand, the authors have not really shown that they are significant at all, at least in the many small rainfall events that the authors describe (34% of rainfall events were smaller than 5 mm depth - see line 271).

Reply: *We found that the line number in this comment ("34% of rainfall events were smaller than 5 mm depth - see line 271") refer to version1 (original version) of this manuscript.*

*The study area is in the south fringe of Mu Us sandy land in North China, and the shrub is distributed sparsely with distinct interspaces. The areal cover provided by shrubs was less than 20%, with the rest of soil being directly exposed to rainfall with no interception (see P.7, Lines 176-178).*

*This study focused on the rainfall partitioning (interception, throughfall, and stemflow). We agree with the reviewer that the small rainfall events are very important for the xerophytic shrubs. We have measured the soil moisture, water isotopes and root systems. In our next research, we will analyze the soil moisture responses to rainfall partitioning and examine how the shrubs actually make use of small amounts of throughfall or stemflow (see P.26, Lines 615-619).*

**6. Comment:**

Minor errors:

line 49: interception loss is not comprised of transpiration
lines 328-329: English expression needs to be improved
line 338: 'slighter' should I think be 'slightly'

Reply: *We found that the line numbers in the comments refer to version1 (original version) of this manuscript. In version2 (first revised version) of this manuscript, we have changed "transpiration" to "evaporation" (see P.3, Line 56).*

*The sentence indicated in the second comment is confusing, and we have deleted it (see P.18, Line 430). "slighter" has been changed to "slightly" (see P.18, Line 442).*